**eLife** RESEARCH ARTICLE

# Interdependence of plasma membrane nanoscale dynamics of a kinase and its cognate substrate underlies *Arabidopsis* response to viral infection

Marie-Dominique Jolivet[1], Anne Flore Deroubaix[1], Marie Boudsocq[2], Nikolaj B Abel[3,4†], Marion Rocher[1], Terezinha Robbe[1], Valérie Wattelet-Boyer[1], Jennifer Huard[1], Dorian Lefebvre[2], Yi-Ju Lu[5], Brad Day[5], Grégoire Saias[1], Jahed Ahmed[1], Valérie Cotelle[6], Nathalie Giovinazzo[7], Jean-Luc Gallois[7], Yasuyuki Yamaji[8], Sylvie German-Retana[9], Julien Gronnier[1,10], Thomas Ott[3,4,11], Sébastien Mongrand[1], Véronique Germain[1]*

[1]Univ. Bordeaux, CNRS, Laboratoire de Biogenèse Membranaire (LBM), Villenave d'Ornon, France; [2]Université Paris-Saclay, CNRS, INRAE, Univ Evry, Université Paris Cité, Institute of Plant Sciences Paris-Saclay (IPS2), Saclay, France; [3]Faculty of Biology, University of Freiburg, Freiburg, Germany; [4]Faculty of Biology, University of Munich (LMU), Munich, Germany; [5]Department of Plant, Soil and Microbial Sciences, Michigan State University, East Lansing, United States; [6]Laboratoire de Recherche en Sciences Végétales (LRSV), Université de Toulouse, CNRS, UPS, Toulouse INP, Toulouse, France; [7]INRAE, GAFL, Montfavet, France; [8]Graduate School of Agricultural and Life Sciences, The University of Tokyo, Tokyo, Japan; [9]UMR 1332 BFP, INRAE Univ. Bordeaux, Bordeaux, France; [10]Center of Plant Molecular Biology (ZMBP), University of Tübingen, Tübingen, Germany; [11]CIBSS – Centre for Integrative Biological Signalling Studies, University of Freiburg, Freiburg, Germany

*For correspondence:
veronique.germain@u-bordeaux.fr

Present address: †Aarhus University, Aarhus, Denmark

## eLife Assessment

The study is considered **important** with **solid** evidence that demonstrates the impact of plasma membrane nano-domains and protein interactions in the plant defence response to viruses. It includes a molecular understanding of the role of a calcium dependent kinase (CPK3) and a remorin protein in the cell-to-cell spread of viruses and cytoskeletal dynamics demonstrating, conclusively, the role of CPK3 with multiple lines of evidence. The work opens avenues to investigate different viruses and other plasma membrane proteins to gain a fuller picture of the involvement of plasmodesmata and other nanodomains in virus spreading.

**Abstract** Plant viruses represent a risk to agricultural production and as only a few treatments exist, it is urgent to identify resistance mechanisms and factors. In plant immunity, plasma membrane (PM)-localized proteins play an essential role in sensing the extracellular threat presented by bacteria, fungi, or herbivores. Viruses are intracellular pathogens and as such the role of the plant PM in detection and resistance against viruses is often overlooked. We investigated the role of the partially PM-bound Calcium-dependent protein kinase 3 (CPK3) in viral infection and we discovered that it displayed a specific ability to hamper viral propagation over CPK isoforms that are involved in immune response to extracellular pathogens. More and more evidence supports that the lateral

organization of PM proteins and lipids underlies signal transduction in plants. We showed here that CPK3 diffusion in the PM is reduced upon activation as well as upon viral infection and that such immobilization depended on its substrate, Remorin (REM1.2), a scaffold protein. Furthermore, we discovered that the viral infection induced a CPK3-dependent increase of REM1.2 PM diffusion. Such interdependence was also observable regarding viral propagation. This study unveils a complex relationship between a kinase and its substrate that contrasts with the commonly described co-stabilisation upon activation while it proposes a PM-based mechanism involved in decreased sensitivity to viral infection in plants.

## Introduction

Viruses are intracellular pathogens, carrying minimal biological material and strictly relying on their host for replication and propagation. They represent a critical threat to both human health and food security. In particular, potexvirus epidemics like the one caused by pepino mosaic virus dramatically affect crop production (*Hanssen and Thomma, 2010*) and the lack of chemical treatments available makes it crucial to develop inventive protective methods. Unlike their animal counterparts, which enter host cells by interacting directly with the plasma membrane (PM), plant viruses have to rely either on mechanical wounding or insect vectors to cross the plant cell wall (*Hanssen and Thomma, 2010*). For this reason, only a few PM-localized proteins were identified as taking part in immunity against viruses (*Zorzatto et al., 2015*; *Ngou et al., 2022*). Among them, members of the REMORIN (REM) protein family were shown to be involved in viral propagation, with varying mechanisms depending on the studied viral genera (*Cheng et al., 2020*; *Fu et al., 2018*; *Ma et al., 2022a*; *Raffaele et al., 2009*; *Rocher et al., 2022*; *Son et al., 2014*). REM proteins are well-known for their heterogeneous distribution at the PM, forming nanodomains (ND), PM nanoscale environments that display a composition different from the surrounding PM (*Gronnier et al., 2018*; *Jaillais and Ott, 2020*, *Jarsch et al., 2014*). Increasing evidence supports the role of ND in signal transduction, with the underlying idea that the local accumulation of proteins allows amplification and specification of the signal (*Jaillais and Ott, 2020*). For example, *Arabidopsis* RHO-OF-PLANT 6 accumulates in distinct ND upon osmotic stress and auxin treatment in a dose-dependent way for the latter (*Platre et al., 2019*). Recently, *Arabidopsis* REM1.2 (later named REM1.2) was demonstrated to form clusters upon exposure to the bacterial effector flg22 to support the condensation of *Arabidopsis* FORMIN 6 and to induce actin cytoskeleton remodeling (*Ma et al., 2022b*). However, unlike the canonical mechanism describing the accumulation of proteins in ND upon stimulation (*Platre et al., 2019*; *Ma et al., 2022b*; *Smokvarska et al., 2020*), we showed previously that *Solanum tuberosum* REM1.3 (StREM1.3) ND were disrupted and the diffusion of individual proteins increased in response to a viral infection (*Perraki et al., 2018*). StREM1.3 lateral organization in lipid bilayers was also shown to be modified upon its phosphorylation status, both in vitro and in vivo (*Perraki et al., 2018*; *Legrand et al., 2023*). The role of such protein dispersion upon stimuli is not understood yet and only a few similar cases are reported in the literature (*Martinière et al., 2019*; *Gournas et al., 2018*) REM1.2 was identified as a substrate of the partially membrane-bound calcium-dependent protein kinase 3 (CPK3) (*Mehlmer et al., 2010*). CPK3 is involved in defense response against herbivores, bacteria, and viruses and was recently proposed to be at cross-roads between pattern-triggered immunity and effector-triggered immunity (*Perraki et al., 2018*; *Kanchiswamy et al., 2010*; *Lu et al., 2020*). We showed previously that transient over-expression of CPK3 in *N. benthamiana* was able to hamper potato virus X (PVX, potexvirus) cell-to-cell propagation (*Perraki et al., 2018*). Although partially PM localized, the role of CPK3 PM localization in immunity was never investigated for any pathogen. As the PVX cannot infect *Arabidopsis thaliana*, the dedicated plant model for genetic studies, we used an alternative virus model able to infect this species, the plantago asiatica mosaic virus (PlAMV) (*Minato et al., 2014*; *Yamaji et al., 2012*) to investigate the role of CPK3 and of its PM localization in potexvirus propagation. We were able to highlight the specific role of CPK3 among other immune-related CPKs *Boudsocq et al., 2010*; *Gao et al., 2013*; *Guzel Deger et al., 2015* in this context. Also, we demonstrated that CPK3 membrane localization was required to hamper viral cell-to-cell propagation, and using single-particle tracking photoactivated light microscopy (spt-PALM), we discovered that CPK3 diffusion was reduced upon activation and viral infection. Interestingly, this reduction of PM diffusion depended on the expression of Group 1 REM while viral-induced REM1.2 increased PM diffusion depended on CPK3. Overall,

our data allowed us to propose a model for a PM-localized mechanism involved in the reduction of potexvirus propagation, which will encourage further exploration of the involvement of the PM in viral immunity.

## Results

### *Arabidopsis thaliana* CPK3 specifically restricts PlAMV propagation

We previously showed that transient overexpression of *Arabidopsis* CPK3 in *N. benthamiana* leaves restricted the propagation of the potexvirus potato virus X (PVX) (*Perraki et al., 2018*). CPKs are encoded by a large gene family of 34 members in *Arabidopsis* (*Yip Delormel and Boudsocq, 2019*). To evaluate the functional redundancy between CPKs regarding viral propagation, a series of *Arabidopsis* lines mutated for CPK1, CPK2, CPK3, CPK5, CPK6, or CPK11, that are involved in plant resistance to various pathogens (*Lu et al., 2020*; *Boudsocq et al., 2010*; *Gao et al., 2013*; *Guzel Deger et al., 2015*; *Dubiella et al., 2013*), were analyzed in a viral propagation assay. Because PVX does not infect *Arabidopsis*, we used instead a binary plasmid encoding for the genomic structure of a GFP-tagged PlAMV (*Minato et al., 2014*; *Yamaji et al., 2012*), a potexvirus that is capable of infecting a wide range of plant hosts, including *Arabidopsis*. Agrobacterium carrying PlAMV-GFP were infiltrated in *A. thaliana* leaves and GFP-fluorescent infection foci were observed 5 d post infiltration (dpi) (*Figure 1—figure supplement 1*). The following combinations of mutants were tested: the *cpk1 cpk2* double mutant (*Gao et al., 2013*), the *cpk5 cpk6* double mutant (*Boudsocq et al., 2010*), the triple mutant *cpk5 cpk6 cpk11* (*Boudsocq et al., 2010*), and the two quadruple mutants *cpk1 cpk2 cpk5 cpk6* (*Gao et al., 2013*) and *cpk3 cpk5 cpk6 cpk11* (*Guzel Deger et al., 2015*). No significant difference of PlAMV-GFP infection foci area was detected between Col-0, *cpk1 cpk2*, *cpk5 cpk6*, *cpk1 cpk2 cpk5 cpk6* and *cpk5 cpk6 cpk11,* demonstrating that CPK1, CPK2, CPK5, CPK6 and CPK11 are not involved in PlAMV propagation (*Figure 1A and B*). However, a 20% increase of PlAMV-GFP infected area was observed in *cpk3 cpk5 cpk6 cpk11* quadruple mutant relative to the control Col-0, which indicates that CPK3 could play a specific role in viral propagation. Since group 1 REMs are known substrates of CPK3 (*Mehlmer et al., 2010*), we checked REM1.2 phosphorylation specificity by the CPK isoforms tested in viral propagation (*Figure 1—figure supplement 2*). CPK1, 2, and 3 displayed the strongest kinase activity on REM1.2 while CPK5, CPK6, and CPK11 displayed a residual activity. In contrast, all six CPKs phosphorylated the generic substrate histone, suggesting some substrate specificity for REM1.2 in vitro. Since CPK1 and CPK2 were described to be mainly localized within the peroxisome (*Dammann et al., 2003*) and endoplasmic reticulum (*Lu and Hrabak, 2002*), respectively, we hypothesize that they likely do not interact in vivo with PM-localized REM1.2 (*Huang et al., 2019*).

To further confirm a role for CPK3 in PlAMV infection, two independent knock-out (KO) lines for CPK3, *cpk3-1* (*Mori et al., 2006*) and *cpk3-2* (*Mehlmer et al., 2010*) along with two independent lines overexpressing CPK3 i.e., Pro35S:CPK3-HA-OE #8.2 and Pro35S:CPK3-HA-OE#16.2 (*Concordet and Haeussler, 2018*; *Figure 1—figure supplement 3*) were infiltrated with PlAMV-GFP. In both *cpk3-1* and *cpk3-2* KO lines, PlAMV-GFP propagation was significantly enhanced (40 to 60% compared with WT Col-0), whereas the over-expression lines Pro35S:CPK3-HA-OE#8.2 and Pro35S:CPK3-HA-OE#16.272 showed 10% and 20% restriction of the foci area, respectively (*Figure 1C*). To assess whether CPK3 regulates viral propagation at the plant level, the propagation of PlAMV-GFP in systemic leaves was assessed in 3-wk-old *cpk3-2* and Pro35S:CPK3-HA-OE#16.2 lines along with Col-0 at 10, 14 and 17 d post-infection (dpi) using a CCD Camera equipped with a GFP filter (*Figure 1—figure supplement 1*). We observed that loss of CPK3 led to an increase of PlAMV-GFP propagation in distal leaves during the course of our assay while the overexpression of CPK3 did not hamper PlAMV-GFP to a greater extent than WT Col-0 (*Figure 1D and E*). This would suggest that CPK3 effect on PlAMV propagation is predominant at the site of infection. For this reason, we concentrated on foci in local leaves for further study.

CPK3 Lysine 107 functions as an ATP binding site and its substitution into methionine abolishes CPK3 activity in vitro (*Lu et al., 2020*). In good agreement, we observed that CPK3$^{K107M}$ can no longer phosphorylate REM1.2 in vitro (*Figure 1F*). To test whether CPK3 kinase activity is required for its function during PlAMV infection, we analyzed the propagation of PlAMV-GFP in complementation lines of *cpk3-2* with WT CPK3 or with CPK3$^{K107M}$. While the WT CPK3 fully complemented *cpk3-2*

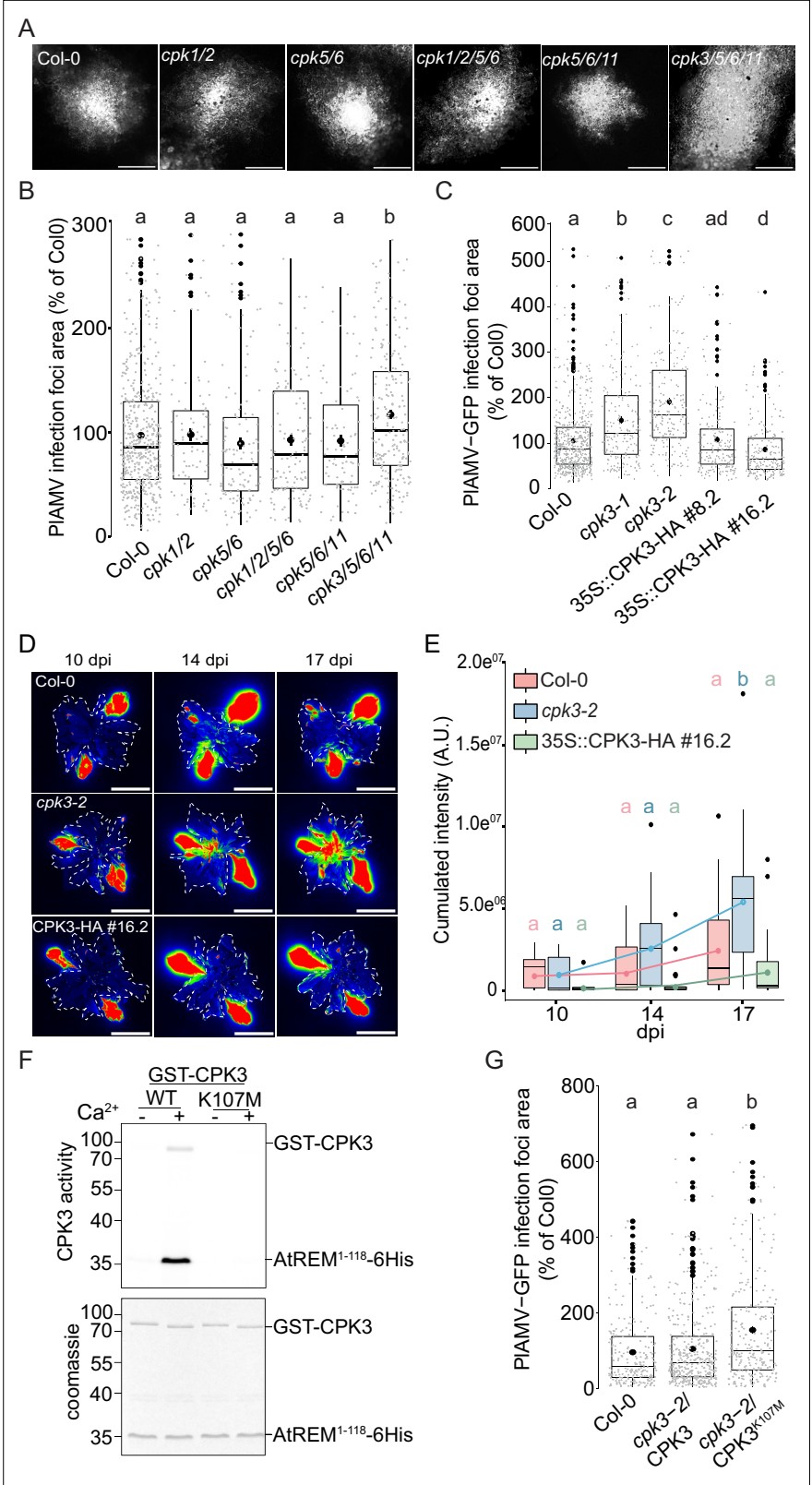

**Figure 1.** *Arabidopsis thaliana* calcium-dependent protein kinase 3 (CPK3) is specifically involved in the restriction of plantago asiatica mosaic virus (PlAMV) cell-to-cell movement. (**A**) Representative images of PlAMV-GFP infection foci at 5 dpi in the different mutant backgrounds. Scale bar = 500 µm. (**B**) Box plots of the mean area of PlAMV-GFP infection foci 5 d after infection in CPK multiple mutant lines, normalized to the mean area

*Figure 1 continued on next page*

*Figure 1 continued*

measured in Col-0. Three independent biological repeats were performed, with at least 47 foci per experiment and per genotype. Significant differences were revealed using a One-Way ANOVA followed by a Tukey's multiple comparison test. Letters are used to discriminate between statistically different conditions (p<0.05). (**C**) Box plots of the mean area of PlAMV-GFP infection foci in *cpk3-1* and *cpk3-2* single mutants and in CPK3 over-expressing lines (Pro35S:CPK3-HA #8.2 and Pro35S:CPK3-HA #16.2), normalized to the mean area measured in Col-0. Three independent biological repeats were performed, with at least 56 foci per experiment and per genotype. Significant differences were revealed using a one-way ANOVA followed by a Tukey's multiple comparison test. Letters are used to discriminate between statistically different conditions (p<0.05). (**D**) Representative images of *A. thaliana* plants infected with PlAMV-GFP and imaged with a CCD Camera from 10 to 17 dpi. The region of interest used for measurement of pixel intensity is circled with a white dotted line. Multicolored scale is used to enhance contrast and ranges from blue (low intensity) to red (high intensity). Scale bar = 4 cm. (**E**) Box plots of the mean cumulated intensity measured in infected leaves in Col-0, *cpk3-2,* and Pro35S:CPK3-HA #16.2 during systemic viral propagation. Two independent experiments were conducted. Statistical differences could be observed between the genotypes and time-points using a Kruskal-Wallis followed by a Dunn's multiple comparison test (p<0.05). For clarity, only the results of the statistical test of the comparison of the different time-points (10, 14, and 17 dpi) within a genotype are displayed and are color-coded depending on the genotype. (**F**) Kinase activity of CPK3 dead variant. Recombinant proteins GST-CPK3 WT and K107M were incubated with REM1.2$^{1\text{-}118}$-6His in kinase reaction buffer in the presence of EGTA (-) or 100 µM free $Ca^{2+}$ (+). Radioactivity is detected on dried gel (upper panel). The protein amount is monitored by Coomassie staining (lower panel). (**G**) Box plots of the mean area of PlAMV-GFP infection foci in *cpk3-2* complemented lines *cpk3-2*/Pro35S:CPK3-myc and *cpk3−2*/Pro35S:CPK3$^{K107M}$-myc. Three independent biological repeats were performed, with at least 51 foci per experiment and per genotype. Significant differences were revealed using a one-way ANOVA followed by a Tukey's multiple comparison test. Letters are used to discriminate between statistically different conditions (p<0.0001).

The online version of this article includes the following source data and figure supplement(s) for figure 1:

**Source data 1.** Original files for western blot analysis displayed in *Figure 1F*.

**Source data 2.** PDF file containing original western blots for *Figure 1F*, indicating the relevant bands and tested conditions.

**Source data 3.** Related to *Figure 1B*.

**Source data 4.** Related to *Figure 1C*.

**Source data 5.** Related to *Figure 1E*.

**Source data 6.** Related to *Figure 1G*.

**Figure supplement 1.** Viral propagation experimental design.

**Figure supplement 2.** Specificity of calcium-dependent protein kinase 3 (CPK) kinase activity towards REM1.2 in vitro.

**Figure supplement 2—source data 1.** Original files for western blot analysis displayed in *Figure 1—figure supplement 2*.

**Figure supplement 2—source data 2.** PDF file containing original western blots for *Figure 1—figure supplement 2*, indicating the relevant bands and tested conditions.

**Figure supplement 3.** Calcium-dependent protein kinase 3 (CPK3) protein levels in knock-out and overexpressing lines.

**Figure supplement 3—source data 1.** Original files for western blot analysis displayed in *Figure 1—figure supplement 3*.

**Figure supplement 3—source data 2.** PDF file containing original western blots for *Figure 1—figure supplement 3*, indicating the relevant bands and tested conditions.

mutant, it was not the case with *cpk3-2*/Pro35S:CPK3$^{K107M}$, which displayed larger infection foci area compared to WT Col-0 (*Figure 1G*).

Taken together, these results demonstrate a specific involvement for CPK3 among other previously described immune-related CPKs in limiting PlAMV infection.

## PlAMV infection induces a decrease in CPK3 PM diffusion

We next wondered whether CPK3 accumulation is regulated at transcriptional and translational levels upon PlAMV infection. RT-qPCR and western blots of CPK3 in Col-0 showed that both transcript and protein levels remained unchanged during PlAMV infection (*Figure 2—figure supplement 1*). CPK3

is partially localized within the PM and is myristoylated at Glycine 2, a modification required for its association with membranes (*Mehlmer et al., 2010*). To test whether CPK3 membrane localization is required to hamper PlAMV propagation, we transformed *cpk3-2* mutant with either ProUbi10:CPK3-mRFP1.2 or ProUbi10:CPK3$^{G2A}$-mRFP1.2 (*Figure 2A*) and tested PlAMV infection. We observed that, in contrary to ProUbi10:CPK3-mRFP1.2, ProUbi10:CPK3$^{G2A}$-mRFP1.2 did not complement *cpk3-2* (*Figure 2B*). These observations indicate that CPK3 association with the PM is required for its function in inhibiting PlAMV propagation. We next analyzed the organization of CPK3 PM pool in absence or presence of PlAMV-GFP using confocal microscopy. Imaging of the surface of *A. thaliana* leaf epidermal cells expressing *CPK3-mRFP1.2* showed that the protein displayed a heterogeneous pattern at the PM in both conditions (*Figure 2C*), although the limitation in lateral resolution of confocal microscopy hindered a more detailed analysis of CPK3 PM organization.

Thus, we used single-particle tracking phospho-activated localization microscopy (sptPALM) which overcomes the diffraction limit of confocal microscopy and allows the analysis of the diffusion and organization of single molecules. We used a translational fusion of CPK3 with the true monomeric photoconvertible fluorescent protein mEOS3.2 (*Zhang et al., 2012*) expressed in stable transgenic *Arabidopsis* lines. We imaged these materials in control and upon PlAMV infection. We tracked single-molecule trajectories (*Figure 2D*) from which CPK3 diffusion coefficient (D) was calculated. We observed that CPK3 proteins were overall mobile in control and infected conditions ($\log(D) > -2$; *Figure 2E*) although CPK3 diffusion was reduced upon PlAMV infection (*Figure 2F*). Analysis of the mean squared displacement (MSD), describing the surface explored by single molecules overtime, showed that CPK3 displayed a more confined behavior during a PlAMV infection than in healthy conditions (*Figure 2G*). Additionally, we performed cluster analysis using Voronoï tessellation, a computation method that segments super-resolution images into polygons based on the local molecule density (*Levet et al., 2015*). Voronoï analysis showed that no difference occurred in CPK3 cluster size or proportion of protein localized in cluster upon viral infection (*Figure 2H–K*). Taken together, these results show that CPK3 diffusion parameters were modified upon PlAMV infection, although the nano-organization of the proteins was maintained.

## Truncation of CPK3 auto-inhibitory domain induces its confinement and accumulation in PM ND

CPK3 bears an auto-inhibitory domain that folds over the kinase domain and inhibits its kinase activity in the absence of calcium (*Sheen, 1996*; *Harper et al., 2004*). The truncation of this domain along with the C-terminal regulatory domain results in a calcium-independent, constitutively active CPK3 (CPK3$^{CA}$) (*Boudsocq et al., 2010*) that is lethal when stably expressed in *Arabidopsis* (*Huimin et al., 2021*). For this reason, CPK3$^{CA}$ was transiently expressed in *N. benthamiana* for further analysis. We observed that although both ProUbi10:CPK3-mRFP1.2 and ProUbi10:CPK3$^{CA}$-mRFP1.2 were partially cytosolic when transiently expressed in *N. benthamiana* (*Figure 3—figure supplement 1*), ProUbi10:CPK3$^{CA}$-mRFP1.2 displayed a PM organization in domains discernable by confocal microscopy (*Figure 3A*).

We analyzed the dynamics of ProUbi10:CPK3-mEOS3.2 and ProUbi10:CPK3$^{CA}$-mEOS3.2 by spt-PALM (*Figure 3B*). We observed that the fraction of immobile ProUbi10:CPK3$^{CA}$-mEOS3.2 molecules ($\log(D) < -2$) is more abundant than for ProUbi10:CPK3-mEOS3.2 in absence of viral infection while no significant difference could be deciphered upon PlAMV infection (*Figure 3C and D*). In addition, MSD analysis showed that the motion of CPK3-mEOS3.2 mobile fraction is less confined than the one of CPK3$^{CA}$-mEOS3.2 and CPK3-mEOS3.2 upon PlAMV infection (*Figure 3E*). Overall, this indicates that the mobile and immobile fraction of CPK3 is affected upon PlAMV infection and upon truncation of its auto-inhibitory domain, respectively.

Cluster analysis of CPK3 was performed using tessellation on the localization data obtained with spt-PALM (*Figure 3F*). Although we did not observe any significant differences in distribution of cluster sizes between all compared conditions (*Figure 3G and H*), CPK3$^{CA}$ displayed a significantly higher proportion of proteins localized in ND compared to CPK3 (*Figure 3I*).

While the mechanism(s) governing the clustering of membrane proteins are not fully described, it is widely accepted that lateral organization involves – to some extent – protein-lipid interactions and lipid-lipid organization (*Gronnier et al., 2018*; *Jaillais and Ott, 2020*; *Sezgin et al., 2017*). We observed that the integrity of ProUbi10:CPK3$^{CA}$-mRFP1.2 organization relied on sterols and

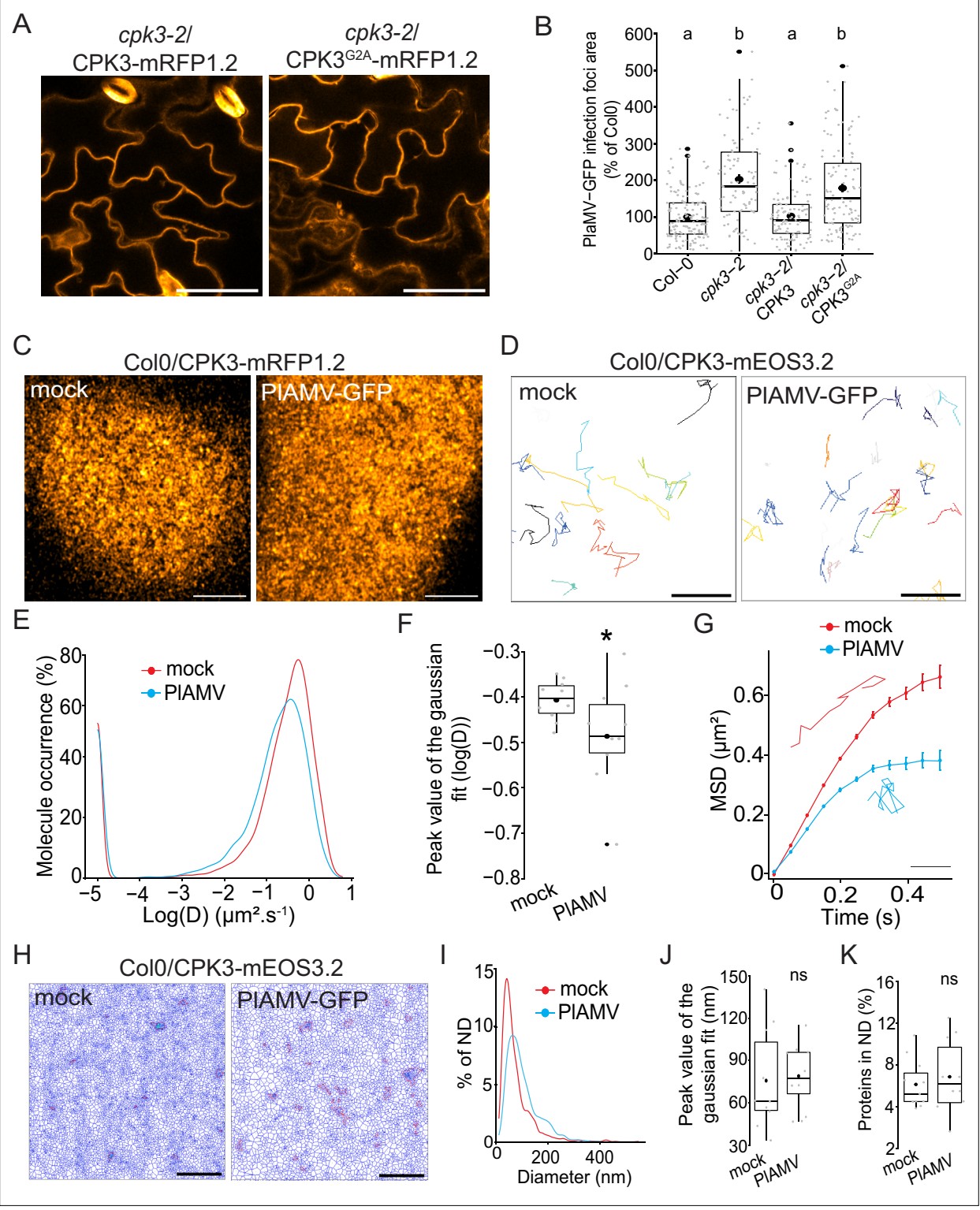

**Figure 2.** Calcium-dependent protein kinase 3 (CPK3) diffusion decreases upon plantago asiatica mosaic virus (PlAMV) infection in *Arabidopsis thaliana*. (**A**) Confocal images of the secant view of *A. thaliana* epidermal cells of the *cpk3-2*/CPK3-mRFP1.2 or *cpk3-2*/ProUbi10:CPK3^G2A-mRFP1.2. Scale bar = 5 μm. (**B**) Box plots of the mean area of PlAMV-GFP infection foci in *cpk3-2* and complemented lines *cpk3-2*/ProUbi10:CPK3-mRFP1.2 and *cpk3-2*/ProUbi10:CPK3^G2A-mRFP1.2. Three independent biological repeats were performed, with at least 32 foci per experiment and per genotype. Significant differences were revealed using a one-way ANOVA followed by a Tukey's multiple comparison test. Letters are used to discriminate between statistically different conditions (p<0.0001). (**C**) Confocal images of the surface view of *A. thaliana* epidermal cells of the Col-0/ProUbi10:CPK3-mRFP1.2, infiltrated

*Figure 2 continued on next page*

*Figure 2 continued*

either with free GFP ('mock') or PlAMV-GFP. Scale bar = 5 µm. (**D**) Representative trajectories of Col-0/CPK3-mEOS3.2 infiltrated either with free GFP ('mock') or PlAMV-GFP; Scale bar = 2 µm. (**E**) Distribution of the diffusion coefficient (D), represented as log(D) for Col-0/ProUbi10:CPK3-mEOS3.2 5 d after infiltration with free GFP ('mock') or PlAMV-GFP. Data were acquired from at least 8086 trajectories obtained in at least 16 cells over the course of three independent experiments. (**F**) Box plot of the mean peak value extracted from the Gaussian fit of log(D) distribution. Significant difference was revealed using a Mann-Whitney test. *p<0.05. (**G**) Mean square displacement (MSD) over time of Col-0/ProUbi10:CPK3-mEOS3.2 5 d after infiltration with free GFP ('mock') or PlAMV-GFP. Representative trajectories extracted from (**D**) illustrate each curve. Scale bar = 1 µm. Data were acquired from at least 16 cells over the course of three independent experiments. (**H**) Voronoi tessellation illustration of Col-0/ProUbi10:CPK3-mEOS3.2 5 d after infiltration with free GFP ('mock') or PlAMV-GFP. ND are circled in red. Scale bar = 2 µm. (**I**) Distribution of the ND diameter of Col-0/ProUbi10:CPK3-mEOS3.2 5 d after infiltration with free GFP ('mock') or PlAMV-GFP. (**J**) Box plot representing the mean peak value of nanodomains (ND) diameter extracted from the Gaussian fit of (**I**). No significant differences were revealed using a Mann-Whitney test. (**K**) Boxplot of the proportion of Col-0/ProUbi10:CPK3-mEOS3.2 detections found in ND 5 d after infiltration with free GFP ('mock') or PlAMV-GFP. No significant differences were revealed using a Mann-Whitney test.

The online version of this article includes the following video, source data, and figure supplement(s) for figure 2:

**Source data 1.** related to *Figure 2B*.

**Source data 2.** related to *Figure 2E*.

**Source data 3.** related to *Figure 2F*.

**Source data 4.** related to *Figure 2G*.

**Source data 5.** related to *Figure 2I*.

**Source data 6.** related to *Figure 2J*.

**Source data 7.** related to *Figure 2K*.

**Figure supplement 1.** Calcium-dependent protein kinase 3 *(CPK3)* transcript and protein levels are not modified upon viral infection.

**Figure supplement 1—source data 1.** Related to *Figure 2—figure supplement 1A*.

**Figure supplement 1—source data 2.** Original files for western blot analysis displayed in *Figure 2—figure supplement 1*.

**Figure supplement 1—source data 3.** Original files for western blot analysis displayed in *Figure 2—figure supplement 1*.

**Figure supplement 2.** Calcium-dependent protein kinase 3 (CPK3) does not accumulate at plasmodesmata during plantago asiatica mosaic virus (PlAMV) infection.

**Figure 2—video 1.** Example of an spt-PALM stream of Col-0/CPK3-mEOS3.2.

https://elifesciences.org/articles/90309/figures#fig2video1

phosphoinositides. Indeed, treatment with fenpropimorph, a well-described inhibitor of sterol biosynthesis (*He et al., 2003*), abolished CPK3$^{CA}$ ND organization (*Figure 3J*), and the co-expression of ProUbi10:CPK3$^{CA}$-mRFP1.2 with the yeast phosphatidylinositol-4-phosphate (PI4P)-specific phosphatase SAC1 targeted to the PM (*Simon et al., 2016*), led to sparser and bigger domains (*Figure 3K*), which suggested that PI4P is not required for CPK3$^{CA}$ ND formation but for its regulation.

All together these observations show that either removal of the auto-inhibitory domain or infection with PlAMV-GFP modifies CPK3 dynamics within the PM; a mechanism which could be at stake for kinase activation.

## PlAMV infection induces an increase in REM1.2 PM diffusion

Group 1 REMs are one of the targets of CPK3 and we previously demonstrated that the restriction of PVX propagation by CPK3 overexpression depended on endogenous group 1 NbREMs (*Perraki et al., 2018*). Four REM isoforms belong to the group 1 in *Arabidopsis*: REM1.1, REM1.2, REM1.3, and REM1.4 (*Raffaele et al., 2007*). *REM1.2* and *REM1.3* are amongst the 10% most abundant transcripts in *Arabidopsis* leaves while REM1.1 was not detected in recently published leaf transcriptomes and proteomes (*Mergner et al., 2020*). Therefore, we focused on the three isoforms REM1.2, REM1.3 and REM1.4. REMs are described as scaffold proteins (*Lefebvre et al., 2010*), for which physiological function depends on the proteins they interact with and their phosphorylation status (*Gouguet et al., 2021*). As recently described in *Abel et al., 2021*, REM1.2 and REM1.3 share 95% of their interactome, suggesting that they are functionally redundant. To address this, we isolated single T-DNA mutants *rem1.2*, *rem1.3* and *rem1.4* (SALK_117637.50.50 .x, SALK_117448.53.95 .x and SALK_073841.47.35, respectively) and crossed them to obtain the double mutant *rem1.2 rem1.3* and the triple mutant *rem1.2 rem1.3 rem1.4* (*Figure 4—figure supplement 1*). We did not notice

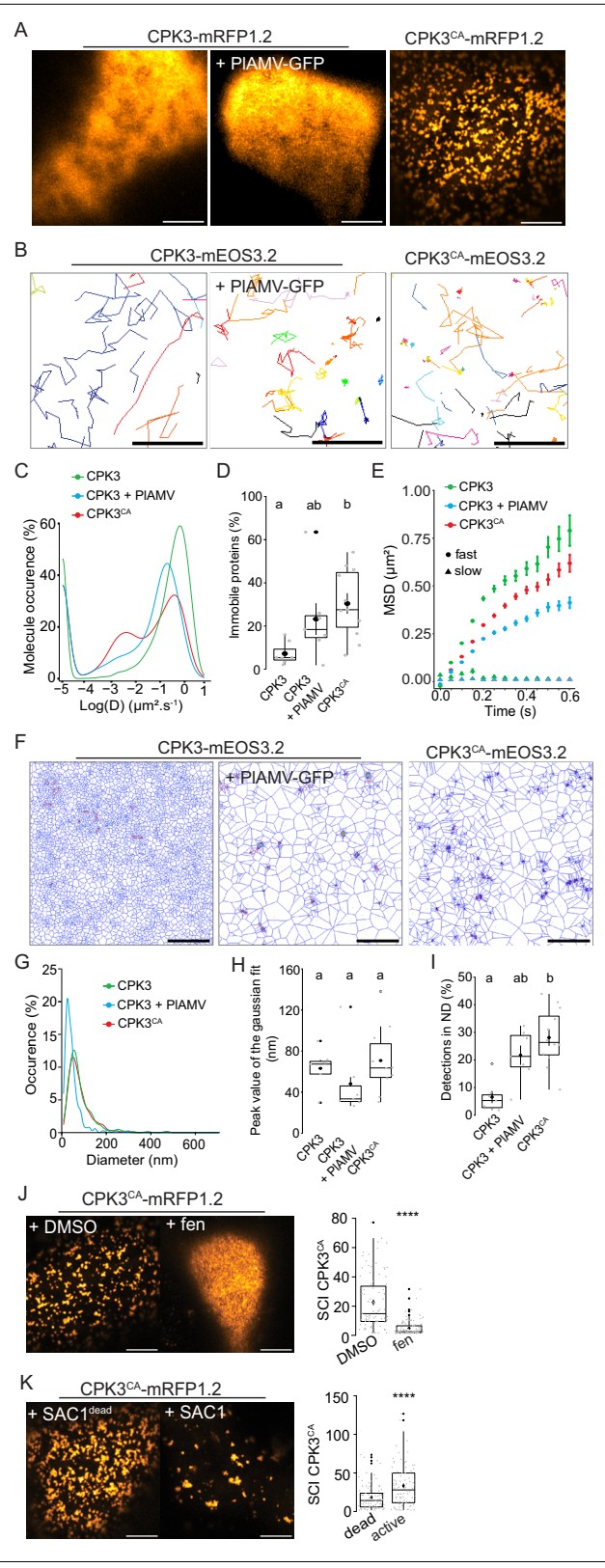

**Figure 3.** Plantago asiatica mosaic virus (PlAMV)-induced activation of calcium-dependent protein kinase 3 (CPK3) in *N. benthamiana* induces its confinement and clustering in plasma membrane (PM) domains. (**A**) Confocal images of the surface view of *N. benthamiana* epidermal cells transiently expressing ProUbi10:CPK3-mRFP1.2, ProUbi10:CPK3-mRFP1.2+PlAMV GFP or ProUbi10:CPK3$^{CA}$-mRFP1.2. Scale bar = 5 µm. (**B**) Representative

*Figure 3 continued on next page*

*Figure 3 continued*

trajectories of ProUbi10:CPK3-mEOS3.2, ProUbi10:CPK3.2-mEOS3.2+PlAMV and ProUbi10:CPK3$^{CA}$-mEOS3.2. Scale bar = 2 μm. (**C**) Distribution of the diffusion coefficient (**D**), represented as log(D) for ProUbi10:CPK3-mEOS3.2, ProUbi10:CPK3-mEOS3.2+PlAMV GFP and ProUbi10:CPK3$^{CA}$-mEOS3.2. Data were acquired from at least 6144 trajectories obtained in at least 15 cells over the course of three independent experiments. (**D**) Box plots of the fraction of immobile proteins (log(D)p<0.005). (**E**) Mean square displacement (MSD) over time of fast (circle) or slow (triangle) diffusing ProUbi10:CPK3-mEOS3.2, ProUbi10:CPK3-mEOS3.2+PlAMV GFP, ProUbi10:CPK3$^{CA}$-mEOS3.2. Standard error is displayed from three independent experiments. (**F**) Voronoï tessellation illustration of ProUbi10:CPK3-mEOS3.2, ProUbi10:CPK3-mEOS3.2+PlAMV GFP and ProUbi10:CPK3$^{CA}$-mEOS3.2. ND are circled in red. Scale bar = 2 μm. (**G**) Distribution of the ND diameter of ProUbi10:CPK3-mEOS3.2, ProUbi10:CPK3-mEOS3.2+PlAMV GFP and ProUbi10:CPK3$^{CA}$-mEOS3.2. (**H**) Box plot representing the mean peak value of ND diameter extracted from the Gaussian fit of (**G**). No significant differences were revealed using a Kruskal-Wallis followed by a Dunn's multiple comparison test.(**I**) Boxplot of the proportion of detections found in ND of ProUbi10:CPK3-mEOS3.2, ProUbi10:CPK3-mEOS3.2+PlAMV GFP and ProUbi10:CPK3$^{CA}$-mEOS3.2. Significant differences were revealed using a Kruskal-Wallis followed by a Dunn's multiple comparison test. Letters are used to discriminate between statistically different conditions (p<0.005). (**J**) Left: Confocal images of the surface view of *N. benthamiana* epidermal cells transiently expressing ProUbi10:CPK3$^{CA}$-mRFP1.2 and infiltrated with either DMSO or 50 μg/mL fenpropimorph. Scale bar = 5 μm; Right: Box plot of the mean spatial clustering index (SCI) of CPK3$^{CA}$. At least three experiments were performed, with at least 10 cells per experiment; statistical significance was determined using a Student *t*-test, ****:p<0.0001. (**K**) Left: Confocal images of the surface view of *N. benthamiana* epidermal cells transiently co-expressing ProUbi10:CPK3$^{CA}$-mRFP1.2 with active or dead SAC1, mutated for its phosphatase activity. Scale bar = 5 μm; Right: Box plot of the mean SCI of CPK3$^{CA}$. At least three experiments were performed, with at least 10 cells per experiment; statistical significance was determined using a Student *t*-test, ****p<0.0001.

The online version of this article includes the following video, source data, and figure supplement(s) for figure 3:

**Source data 1.** Related to *Figure 3C*.

**Source data 2.** Related to *Figure 3D and E*.

**Source data 3.** Related to *Figure 3G*.

**Source data 4.** Related to *Figure 3H*.

**Source data 5.** Related to *Figure 3I*.

**Source data 6.** Related to *Figure 3J*.

**Source data 7.** Related to *Figure 3K*.

**Figure supplement 1.** Calcium-dependent protein kinase 3 (CPK3) and CPK3$^{CA}$ display a similar subcellular localization.

**Figure 3—video 1.** Example of an spt-PALM stream of CPK3-mEOS3.2 transiently expressed in *N. benthamiana*. https://elifesciences.org/articles/90309/figures#fig3video1

**Figure supplement 2.** Calcium-dependent protein kinase 3 (CPK3) and CPK3CA display a similar subcellular localization.

any obvious defects in the growth and development of seedlings and adult plants for the single, double, and triple mutants, when grown under our conditions (*Figure 4—figure supplement 2*). No difference in PlAMV-GFP propagation could be observed in the single mutants, compared to Col-0 (*Figure 4A*). However, the double mutant *rem1.2 rem1.3* showed a significant increase of infection foci area compared to Col-0, which was further enhanced in *rem1.2 rem1.3 rem1.4* triple KO mutant. Such additive effect of multiple mutations shows that REM1.2, REM1.3, and REM1.4 are functionally redundant regarding PlAMV cell-to-cell propagation. Finally, PlAMV-GFP systemic propagation was followed in whole plants every 3–4 d from 10 to 17 dpi, and *rem1.2 rem1.3 rem1.4* displayed an increased infection surface of systemic leaves compared to Col-0 (*Figure 4B and C*), suggesting that group 1 REMs are involved in both local and systemic propagation of PlAMV-GFP.

Given their role in cell-to-cell viral movement, we checked whether group 1 REM expression was modified upon infection. RT-qPCR and western blots showed that neither transcripts nor protein levels were modified upon PlAMV infection (*Figure 4—figure supplement 3*). Since REM1.2 and REM1.3 share a large part of their interactome and show functional redundancy regarding PlAMV infection, we decided to focus on REM1.2 for further investigations. Confocal imaging of the surface of epidermal cells of Col-0/ProUbi10:mRFP1.2-REM1.2 showed a rather heterogeneous distribution at

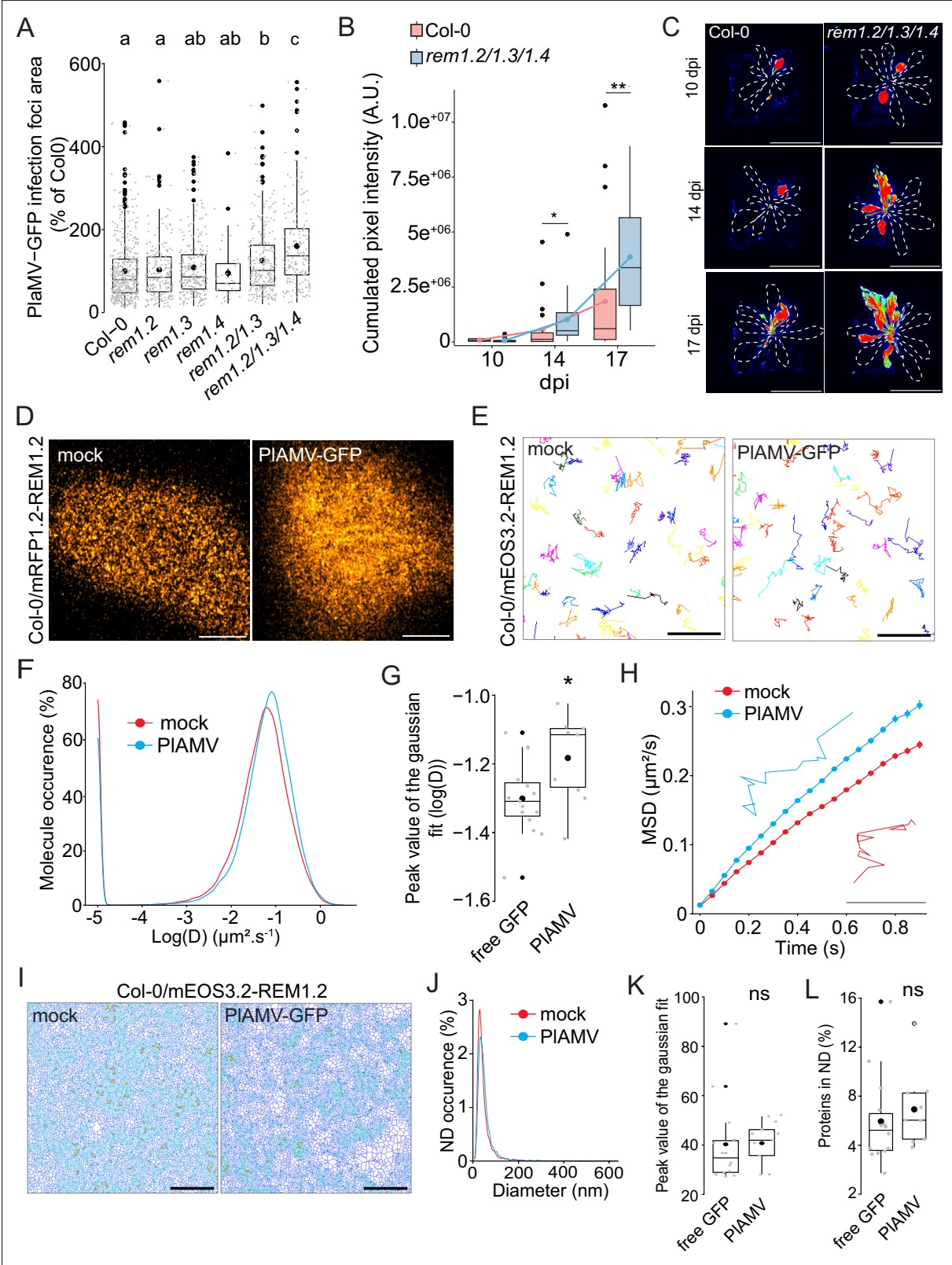

**Figure 4.** Group 1 REMORIN (REM) hampers plantago asiatica mosaic virus (PlAMV)-GFP cell-to-cell propagation and REM1.2 diffusion increases upon infection. (**A**) Box plots of the mean area of PlAMV-GFP infection foci in *rem1.2, rem1.3, rem1.*4 single mutants along with *rem1.2 rem1.3* double mutant and *rem1.2 rem1.3 rem1.4* triple mutant. Three independent biological repeats were performed, with at least 36 foci per experiment and per genotype. Significant differences were revealed using a one-way ANOVA followed by a Tukey's multiple comparison test. Letters are used to discriminate between

*Figure 4 continued on next page*

*Figure 4 continued*

statistically different conditions (p<0.05). (**B**) Box plots of the mean cumulated intensity measured in infected leaves in Col-0 and *rem1.2 rem1.3 rem1.4* during systemic viral propagation. Two independent experiments were conducted. Statistical significance of the difference between Col-0 and *rem1.2 rem1.3 rem1.4* at each time point was assessed using a Mann-Whitney test. *p<0.05, **p<0.01. (**C**) Representative images of *A. thaliana* plants infected with PlAMV-GFP and imaged with a CCD Camera from 10 to 17 dpi. Systemic leaves are circled with a white dotted line. Multicolored scale is used to enhance contrast and ranges from blue (low intensity) to red (high intensity). Scale bar = 4 cm. (**D**) Confocal images of the surface view of *A. thaliana* epidermal cells of the Col-0/ProUbi10:mRFP1.2-REM1.2, infiltrated either with free GFP ('mock') or PlAMV-GFP. Scale bar = 5 μm (**E**) Representative trajectories of Col-0/ProUbi10:mEOS3.2-REM1.2 5 d after infiltration with free GFP ('mock') or PlAMV-GFP. Scale bar = 2 μm. (**F**) Distribution of the diffusion coefficient (**D**), represented as log(D) for Col-0/ProUbi10:mEOS3.2-REM1.2 5 d after infiltration with free GFP ('mock') or PlAMV-GFP. Data were acquired from at least 28638 trajectories obtained from at least 16 cells over the course of three independent experiments. (**G**) Box plot of the mean peak value extracted from the Gaussian fit of log(D) distribution. Significant difference was revealed using a Mann-Whitney test. *p<0.05. (**H**) Mean square displacement (MSD) over time of Col-0/ProUbi10:mEOS3.2-REM1.2 infiltrated either with free GFP ('mock') or PlAMV-GFP. Representative trajectories extracted from (**E**) illustrate each curve. Scale bar = 1 μm. (**I**) Voronoi tessellation illustration of Col-0/ProUbi10:mEOS3.2-REM1.2 5 d after infiltration with free GFP ('mock') or PlAMV-GFP. ND are circled in red. Scale bar = 2 μm (**J**) Distribution of the nanodomains (ND) diameter of Col-0/ProUbi10:mEOS3.2-REM1.2 5 d after infiltration with free GFP ('mock') or PlAMV-GFP. (**K**) Box plot representing the mean peak value of ND diameter extracted from the Gaussian fit of (**J**). No significant difference was revealed using a Mann-Whitney test. (**L**) Boxplot of the proportion of Col-0/ProUbi10:mEOS3.2-REM1.2 detections found in ND 5 d after infiltration with free GFP ('mock') or PlAMV-GFP. No significant difference was revealed using a Mann-Whitney test.

The online version of this article includes the following video, source data, and figure supplement(s) for figure 4:

**Source data 1.** Related to *Figure 4A*.

**Source data 2.** Related to *Figure 4B*.

**Source data 3.** Related to *Figure 4F*.

**Source data 4.** Related to *Figure 4G*.

**Source data 5.** Related to *Figure 4H*.

**Source data 6.** Related to *Figure 4J*.

**Source data 7.** Related to *Figure 4K*.

**Source data 8.** Related to *Figure 4L*.

**Figure supplement 1.** Group 1 REMORIN (REM) single, double, and triple knock-out lines.

**Figure supplement 1—source data 1.** Original files for western blot analysis displayed in *Figure 4—figure supplement 1*.

**Figure supplement 1—source data 2.** PDF file containing original western blots for *Figure 4—figure supplement 1*, indicating the relevant bands and tested conditions.

**Figure supplement 2.** Group 1 REMORINs (REMs) single and multiple knock-out lines do not display any obvious developmental phenotype.

**Figure supplement 3.** Group 1 REMORINs (REMs) transcript and protein levels are not modified during a plantago asiatica mosaic virus (PlAMV) infection.

**Figure supplement 3—source data 1.** Related to *Figure 4—figure supplement 3A*.

**Figure supplement 3—source data 2.** Original files for western blot analysis displayed in *Figure 4—figure supplement 3*.

**Figure supplement 3—source data 3.** PDF file containing original western blots for *Figure 4—figure supplement 3*, indicating the relevant bands and tested conditions.

**Figure supplement 4.** REM1.2 does not accumulate at plasmodesmata during plantago asiatica mosaic virus (PlAMV) infection.

**Figure 4—video 1.** Example of an spt-PALM stream of Col-0/mEOS3.2-REM1.2.

https://elifesciences.org/articles/90309/figures#fig4video1

---

the PM, although less striking than previously described when observed in root (*Huang et al., 2019*; *Figure 4D*). REM1.2 was next fused to mEOS3.2 and stably expressed in Col-0 to conduct spt-PALM (*Figure 4E*). The diffusion coefficient of ProUbi10:mEOS3.2-REM1.2 was significantly increased upon PlAMV infection (*Figure 4F and G*). Moreover, its MSD was increased to a similar extent as what was previously observed with StREM1.3 during a PVX infection (*Perraki et al., 2018*; *Figure 4H*), although REM1.2 is overall more mobile than StREM1.3. Tessellation analysis of protein localization did not show any difference in ND organization, whether in size or regarding the enrichment of proteins in ND (*Figure 4I–L*).

Taken together, these results show that group 1 REMs are functionally redundant regarding their ability to hamper PlAMV propagation. PlAMV infection promoted an increased diffusion of REM1.2,

in the same way as PVX did with StREM1.3. The conservation of such mechanism between plants of different families is indicative of its physiological importance.

## PlAMV-induced changes in REM1.2 and CPK3 plasma membrane dynamics are interdependent

Group 1 REMs from *A. thaliana* were previously identified as in vitro substrates of CPK3 (*Mehlmer et al., 2010*). Moreover, untargeted immunoprecipitation experiments coupled to mass spectrometry identified CPK3 as an interactor of REM1.2 in *A. thaliana* (*Abel et al., 2021*). We wanted to assess the functional link between CPK3 and group 1 REM in potexvirus propagation by knocking out CPK3 into the *rem1.2 rem1.3 rem1.4* mutant background. We isolated two independent CRISPR-generated *rem1.2 rem1.3 rem1.4 cpk3* #1 and *rem1.2 rem1.3 rem1.4 cpk3* #2 quadruple mutants (*Figure 5—figure supplement 1*). We did not observe any developmental defect in these lines when grown under controlled conditions (*Figure 5—figure supplement 2*). The analysis of PlAMV-GFP propagation showed that no significant additive effect could be observed between the quadruple mutant lines, *cpk3-2* and *rem1.2 rem1.3 rem1.4* (*Figure 5A and B*). This indicates that group 1 REMs and CPK3 function in the same signaling pathway to inhibit PlAMV propagation.

We wanted to know whether the increased diffusion of REM1.2 observed on PlAMV infection was dependent on CPK3. Using spt-PALM, we obtained the diffusion parameters of ProUbi10:REM1.2-mEOS3.2 expressed in the *cpk3-2* mutant background. Strikingly, we observed that both the diffusion coefficient and MSD of REM1.2 were not anymore affected during a viral infection (*Figure 5C–F*), showing that REM1.2 PM lateral diffusion upon PlAMV infection depends on CPK3. In a similar manner as in Col-0, REM1.2 clustering upon PlAMV infection in *cpk3-2* background did not display any difference to the mock-infected plants (*Figure 5—figure supplement 3*).

Moreover, we wondered whether the reciprocal effect was true for the diffusion of CPK3 in the absence of group 1 REMs. Similarly, the diffusion coefficient and the MSD of mEOS3.2-CPK3 in *rem1.2 rem1.3 rem1.4* triple KO background remained the same during an infection compared to control condition (*Figure 5G–J*), unlike what was observed in a Col-0 background (*Figure 2D–G*). This result indicated that the confinement of CPK3 proteins upon viral infection depended on the presence of group 1 REMs. Moreover, contrarily to the Col-0 background, the *rem1.2 rem1.3 rem1.4* displayed reduced protein concentration upon PlAMV infection (*Figure 5—figure supplement 3*). This showed that group 1 REMs might play a role in CPK3 domain regulation upon viral infection.

As CPK3[CA] was shown to inhibit potexvirus propagation more efficiently than the full-length protein (*Perraki et al., 2018*), we tested CPK3[CA] ability to alter REM diffusion (*Figure 5—figure supplement 4*). Upon transient co-expression with CPK3[CA], REM diffusion was significantly increased. As shown in *Figure 2B*, CPK membrane anchor is crucial for its role in viral propagation. We tested if CPK3[CA-G2A] was still able to hinder REM diffusion, which was not the case. All these data support a specific role of PM-bound CPK3 in REM increased diffusion upon viral infection.

We then investigated whether CPK3 and REM would colocalize in absence or presence of the virus. Using confocal microscopy, we showed that they randomly colocalized in both situations (*Figure 5—figure supplement 5*), the interaction between the kinase and its substrate probably occurring in a narrow spatiotemporal window.

Taken all together, those results show a strong inter-dependence of group 1 REMs and CPK3 both in their physiological function and in their PM lateral diffusion upon PlAMV infection.

## Discussion

### CPK3 specific role in viral immunity is supported by its PM organization

Although calcium-mediated signaling is suspected to be involved in viral immunity, only a few calcium-modulated proteins are described as playing a role in viral propagation (*Perraki et al., 2018*; *Wang et al., 2021*). We showed here the crucial role of CPK3 over other immunity-related CPK isoforms *Boudsocq et al., 2010*; *Gao et al., 2013* by a reverse genetic approach in *Arabidopsis*. The precise role of CPK3 in viral immunity remains to be determined. CPK3 phosphorylates actin depolymerization factors to modulate the actin cytoskeleton (*Lu et al., 2020*; *Dong and Hong, 2013*), a key player in host-pathogen interaction (*Wang et al., 2022*). In particular, potexviruses induce the remodeling of the actin cytoskeleton to organize the key steps of their cycle, whether it is replication, intra-cellular

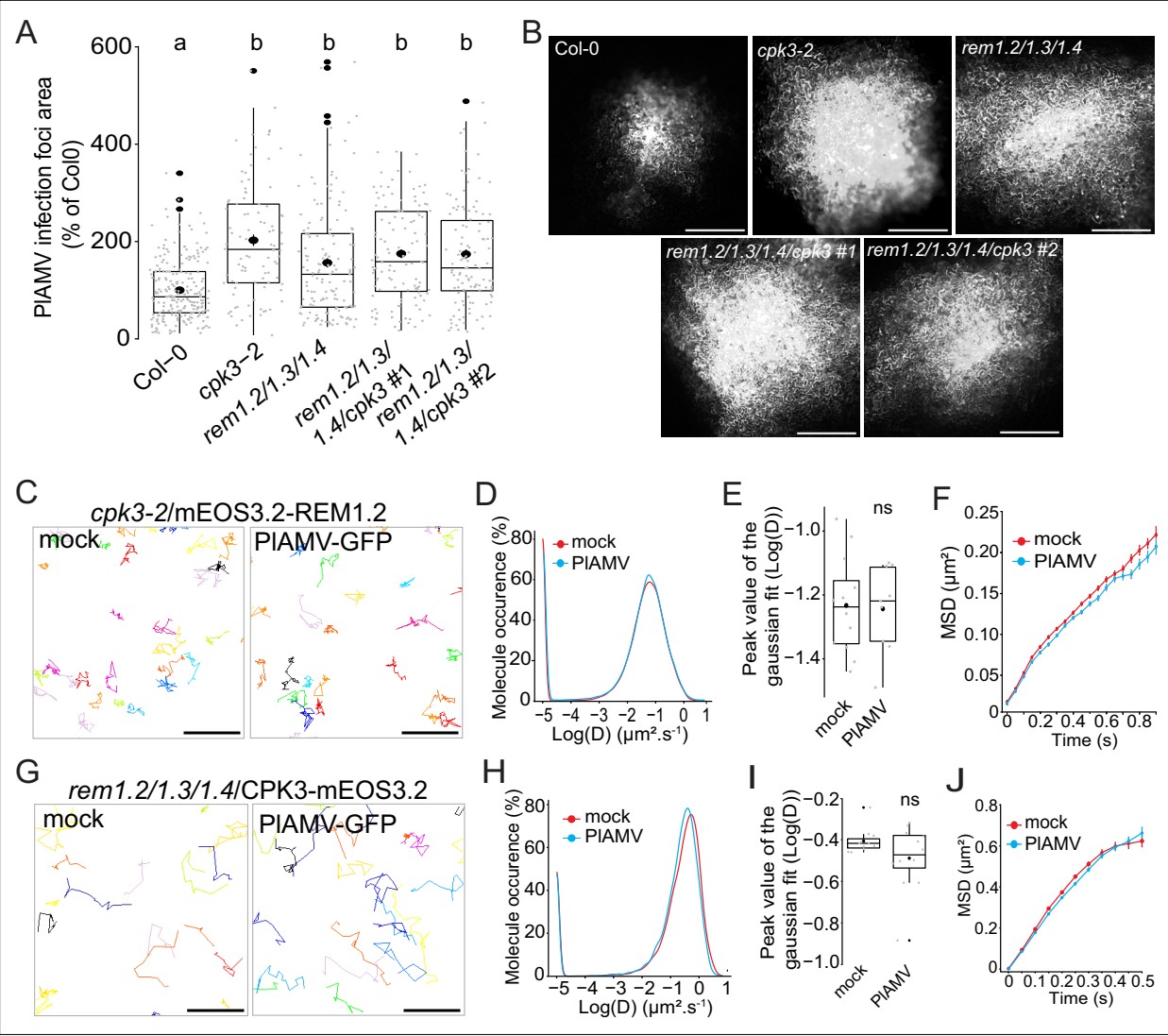

**Figure 5.** Group 1 REMORINs (REMs) and calcium-dependent protein kinase 3 (CPK3) are in the same functional pathway and regulate each other plasma membrane (PM) diffusion upon plantago asiatica mosaic virus (PlAMV) infection. (**A**) Box plots of the mean area of PlAMV-GFP infection foci in *cpk3-2*, *rem1.2 rem1.3 rem1.4* triple mutant and *rem1.2 rem1.3 rem1.4/cpk3 #1 and #2* quadruple mutants. Three independent biological repeats were performed, with at least 23 foci per experiment and per genotype. Significant differences were revealed using a one-way ANOVA followed by a Tukey's multiple comparison test. Letters are used to discriminate between statistically different conditions (p<0.0001). (**B**) Representative images of PlAMV-GFP infection foci at 5 dpi in the different mutant backgrounds. Scale bar = 500 μm.(**C**) Representative trajectories of *cpk3-2*/ProUbi10:mEOS3.2-REM1.2 5 d after infiltration with either free GFP ('mock') or PlAMV-GFP. Scale bar = 2 μm. (**D**) Distribution of the diffusion coefficient (**D**), represented as log(D) for *cpk3-2*/ProUbi10:mEOS3.2-REM1.2 5 d after infiltration with either free GFP ('mock') or PlAMV-GFP. Data were acquired from at least 20462 trajectories obtained from at least 11 cells over the course of three independent experiments. (**E**) Box plot representing the mean peak value extracted from the Gaussian fit of the distribution of the diffusion coefficient (**D**) represented in (**D**). No significant difference was revealed using a Mann-Whitney test. (**F**) Mean square displacement (MSD) over time of *cpk3-2*/ProUbi10:mEOS3.2-REM1.2 5 d after infiltration with free GFP ('mock') or PlAMV-GFP. (**G**) Representative trajectories of *rem1.2 rem1.3 rem1.4*/ProUbi10:CPK3-mEOS3.2 5 d after infiltration with free GFP ('mock') or PlAMV-GFP. Scale bar = 2 μm. (**H**) Distribution of the diffusion coefficient (**D**), represented as log(D) for *rem1.2 rem1.3 rem1.4*/ProUbi10:CPK3-mEOS3.2 5 d after infiltration with free GFP ('mock') or PlAMV-GFP. Data were acquired from at least 11724 trajectories obtained from at least 10 cells over the course of three independent experiments. (**I**) Box plot representing the mean peak value extracted from the Gaussian fit of the distribution of the diffusion coefficient (**D**) represented in (**H**). No significant difference was revealed using a Mann-Whitney test. (**J**) Mean square displacement (MSD) over time of *rem1.2 rem1.3 rem1.4*/ProUbi10:CPK3-mEOS3.2 5 d after infiltration with free GFP ('mock') or PlAMV-GFP.

The online version of this article includes the following video, source data, and figure supplement(s) for figure 5:

**Source data 1.** Related to *Figure 5A*.

**Source data 2.** Related to *Figure 5D*.

*Figure 5 continued on next page*

*Figure 5 continued*

**Source data 3.** Related to *Figure 5E*.

**Source data 4.** Related to *Figure 5F*.

**Source data 5.** Related to *Figure 5H*.

**Source data 6.** Related to *Figure 5I*.

**Source data 7.** Related to *Figure 5J*.

**Figure supplement 1.** CRISPR-mediated knock-out of cpk3 in *rem1.2 rem1.3 rem1.4* background.

**Figure supplement 1—source data 1.** Original files for western blot analysis displayed in *Figure 5—figure supplement 1*.

**Figure supplement 1—source data 2.** PDF file containing original western blots for *Figure 5—figure supplement 1*, indicating the relevant bands and tested conditions.

**Figure supplement 2.** *rem1.2 rem1.3 rem1.4 cpk3* quadruple knock-out lines do not display any obvious developmental phenotype.

**Figure supplement 3.** calcium-dependent protein kinase 3 (CPK3) and group1 REMORIN (REM) role in each other plasma membrane (PM) nano-organization during plantago asiatica mosaic virus (PlAMV) infection.

**Figure supplement 3—source data 1.** Related to *Figure 5—figure supplement 3B*.

**Figure supplement 3—source data 2.** Related to *Figure 5—figure supplement 3C*.

**Figure supplement 3—source data 3.** Related to *Figure 5—figure supplement 3D*.

**Figure supplement 3—source data 4.** Related to *Figure 5—figure supplement 3F*.

**Figure supplement 3—source data 5.** Related to *Figure 5—figure supplement 3G*.

**Figure supplement 3—source data 6.** Related to *Figure 5—figure supplement 3H*.

**Figure supplement 4.** REM1.2 diffusion is increased upon co-expression with CPK3$^{CA}$ but not with CPK3$^{CA}$-G2A.

**Figure supplement 4—source data 1.** Related to *Figure 5—figure supplement 4B*.

**Figure supplement 5.** Plantago asiatica mosaic virus (PlAMV) infection does not modify calcium-dependent protein kinase 3 (CPK3) and group1 REMORIN (REM) colocalization coefficient.

**Figure 5—video 1.** Example of an spt-PALM stream of mEOS3.2-REM1.2 transiently expressed in *N. benthamiana*.
https://elifesciences.org/articles/90309/figures#fig5video1

movement, or cell-to-cell propagation (*Harries et al., 2009*; *Tilsner et al., 2012*). PlAMV replication and/or movement could be affected by CPK3-mediated alteration of the cytoskeleton mesh.

The specificity of a calcium-dependent kinase in a given biological process is determined by its expression pattern, subcellular localization and substrate specificity (*Yip Delormel and Boudsocq, 2019*, Köster et al., 2022) Controlled subcellular localization ensures proximity with either stimulus or substrate and here we showed that, similarly to *Arabidopsis* CPK6 *Saito et al., 2018* and *Solanum tuberosum* CPK5 (*Asai et al., 2013*), the disruption of CPK3 membrane anchor led to a loss-of-function phenotype (*Figure 2B*). Interestingly, we observed a reduction in CPK3 PM diffusion upon PlAMV infection, suggesting that not only membrane localization but also protein organization at the PM is important during viral immunity. Moreover, PlAMV-induced CPK3 PM confinement was reminiscent of the diffusion parameters displayed by CPK3$^{CA}$, hinting that viral infection, kinase activation and lateral diffusion are linked. However, it is necessary to remain careful as a truncated protein deprived of its auto-inhibitory domain does not reflect the controlled and context-dependent activation of CPK3. Indeed, stable expression of CPK3$^{CA}$ was previously shown to be lethal (*Huimin et al., 2021*). CPK3$^{CA}$ ever-activated state might lead to stable or unspecific interaction with protein partners along with erratic phosphorylation of substrates, which could explain the PM domains formed by CPK3$^{CA}$ at the PM.

CPK3 nanoscale dynamics upon viral infection might offer another layer of specificity to convey the appropriate response to a given stimulus by ensuring proximity with specific regulators or substrates. It would be interesting to explore whether the reduction of CPK3 diffusion observed upon PlAMV infection is specific to this virus or if it can be extended to other viral species, genera, and even pathogens. Indeed, it was recently shown that CPK3 transcription was enhanced upon infection by viruses from different genera (*Valmonte-Cortes et al., 2022*). Moreover, CPK3 regulates herbivore responses by phosphorylating transcription factors (*Kanchiswamy et al., 2010*), is activated by flg22 in protoplasts (*Mehlmer et al., 2010*), and is proposed to be the target of a bacterial effector to disrupt immune response (*Lu et al., 2020*). Finally, the diffusion and clustering of other PM-localized

CPKs could be investigated as no experimental data exist yet regarding their PM nano-organization. It would be especially relevant to describe these parameters for the CPK isoforms phosphorylating ND-organized NADPH oxidases (*Smokvarska et al., 2020*; *Gao et al., 2013*; *Dubiella et al., 2013*).

## Potential functions for REM1.2 increased diffusion upon PlAMV

REM proteins display a wide range of physiological functions and are proposed to function as scaffold proteins (*Lefebvre et al., 2010*; *Gouguet et al., 2021*; *Su et al., 2023*). Group 1 REMs have been shown to be involved in viral immunity, playing apparent contradictory roles depending on the virus genera (*Rocher et al., 2022*; *Gouguet et al., 2021*; *Perraki et al., 2012*). Herein, we show through a combination of genetic and biochemical analysis that the three most expressed isoforms of group 1 *Arabidopsis* REMs were functionally redundant in inhibiting PlAMV propagation (*Figure 4A*). REMs are anchored at the PM through their C-terminal sequence (*Gouguet et al., 2021*; *Perraki et al., 2012*; *Gronnier et al., 2017*; *Konrad et al., 2014*) but despite displaying a similar PM attachment, potato StREM1.3 is static and forms well-defined membrane compartments (*Gronnier et al., 2017*) while *Arabidopsis* REM1.2 appeared mobile in leaves, with small and potentially short-lived membrane domains of around 70 nm (*Figure 4F–J*). Beyond clear organizational differences, both REM1.2 and StREM1.3 showed an increased mobility upon viral infection in *N. benthamiana* and *Arabidopsis* (*Perraki et al., 2018*; *Figure 4E–H*), which is at contrast with the canonical model linking protein activation with its stabilization into ND (*Jaillais and Ott, 2020*; *Smokvarska et al., 2021*). Particularly, REM1.2 was recently shown to form ND upon elicitation by a bacterial effector (*Ma et al., 2022b*) or upon exposition to bacterial membrane structures (*Tran et al., 2020*). Conservation across plant and virus species of such increased PM diffusion of group 1 REMs indicates that this specific mechanism is crucial for plant response to potexviruses, although its role remains to be deciphered. It was suggested, in the context of tobacco rattle virus infection, that REM1.2 clusters led to an increased lipid order of the PM and a morphological modification of plasmodesmata (PD), inducing a decrease in PD permeability (*Huang et al., 2019*). As REM1.2 does not accumulate at PD upon PlAMV infection (*Figure 4—figure supplement 4*), it might indirectly influence PD permeability through mechanisms yet to be discovered. We previously showed that StREM1.3 modulated the formation of lipid phases in vitro (*Su et al., 2020*) while *Medicago truncatula* SYMREM1 was recently shown to stabilize membrane topology changes in protoplasts (*Su et al., 2020*). Therefore, REM1.2 increased diffusion upon viral infection might alter lipid organization, with consequences on PM and PD-localized proteins (*Grison et al., 2019*). Based on REM's putative scaffolding role (*Lefebvre et al., 2010*; *Jarsch and Ott, 2011*), its increased PM diffusion might modify the stability of REM-supported complexes and induce subsequent signaling, similarly to the way *Oryza sativa* REM4.1 orchestrates the balance between abscisic acid and brassinosteroid pathways by interacting with different protein kinases (*Gui et al., 2016*).

## A PM-localized mechanism involved in hampering viral propagation

Increasing evidence supports the role of PM proteins in plant antiviral mechanism (*Teixeira et al., 2019*). However, while the nano-organization of PM proteins involved in bacterial or fungal immunity begins to be addressed (*Ma et al., 2022b*; *Tran et al., 2020*; *Liang et al., 2018*; *Gronnier et al., 2022*), there is still only scarce information in a viral infection context. Our observations pointed towards an interdependence of REM and CPK3 in their PM diffusion upon PlAMV infection. In particular, CPK3-dependent REM increased diffusion in a potexvirus infection is consistent with our previously reported PM diffusion increase of StREM1.3 phosphomimetic mutant, reminiscent of StREM1.3 behavior upon viral infection (*Perraki et al., 2018*). Moreover, we recently published that in vitro CPK3-phosphorylated StREM1.3 presented disrupted domains on model membranes compared to the non-phosphorylated protein (*Legrand et al., 2023*). The data presented in this paper further support the predominant role of CPK3 in viral-induced REM1.2 diffusion (*Figure 5C–F*). Since the actin cytoskeleton was shown to favor nanometric scale ND including those of group 1 REM (*Szymanski et al., 2015*), CPK3 mediated regulation of the actin cytoskeleton could regulate REM1.2 PM nanoscale organization. We also discovered the essential role of group 1 REMs in the lateral organization of CPK3 upon viral infection (*Figure 5G–J*), which is further supported by the difference between CPK3 clustering parameters when expressed in *N. benthamiana* or in *Arabidopsis*: CPK3 was enriched in ND upon PlAMV infection when transiently expressed in *N. benthamiana* but not in *Arabidopsis* (*Figures 2H, K, 3F and I*). The discrepancy between both species could be linked

to *N. benthamiana* REM1 ability to form stable ND, discernable by confocal microscopy (*Fu et al., 2018*) while we observed small and unstable domains of REM1.2 in *Arabidopsis* leaves (*Figure 4I–K*). Moreover, CPK3 ND organization was disrupted in *rem1.2 rem1.3 rem1.4* triple KO background upon PlAMV infection (*Figure 5—figure supplement 3H*), indicating that group1 REMs are crucial for the spatial organization of CPK3 during a viral infection. Furthermore, the sensitivity of CPK3[CA] domains to the same lipid inhibitors as StREM1.3 (*Gronnier et al., 2017*; *Figure 3J and K*) hints that REM1.2 lipid environment might be required for CPK3 nanoscale organization.

Although the data obtained here do not allow us to determine the sequence of events orchestrating REM1.2 and CPK3 dynamic interaction, hypotheses can be formulated. REM and CPK3 interact in the absence of any stimulus (*Perraki et al., 2018*; *Abel et al., 2021*) but as they display drastically different diffusion parameters (*Figures 2E and 4F*), likely only a small part of proteins interact at basal state. PlAMV infection induces a REM-dependent confinement of CPK3 (*Figures 2 and 5*). Whether CPK3 confinement is concomitant to its activation remains to be confirmed. Such modification of CPK3 mobility might result in an increase probability of interaction with REM1.2, which might lead to increased phosphorylation events and a subsequent increase in mobility of REM1.2. In addition to the above-mentioned putative roles of REM's increased diffusion, such 'kiss-and-go' mechanism between a kinase and its substrate might also be considered as a negative feedback loop to hamper constitutive activation of the system. It was recently shown that the PM organization of a receptor-like kinase and its co-receptor relied on the expression of a scaffold protein (*Gronnier et al., 2022*; *Stegmann et al., 2017*), REM might go beyond the role of substrate and be essential to maintain the required PM environment of its cognate kinase to ensure proper signal transduction. Although the complexity of potexvirus lifecycle as well as technical limitations impair a thorough challenge of this hypothesis, our data support the significance of fine PM compartmentalization and spatio-temporal dynamics in signal transduction upon viral infection in plants while it explores new concepts underlying plant kinases PM organization.

# Materials and methods

## Plant culture

*Nicotiana benthamiana* plants were cultivated in controlled conditions (16 hr photoperiod, 25 °C). Proteins were transiently expressed via *Agrobacterium tumefaciens* as previously described (*Gronnier et al., 2017*). The agrobacteria GV3101 strain was cultured at 28 °C on appropriate selective medium depending on constructs carried. Plants were observed between 2 and 5 d after infiltration depending on experiments.

Sterilized *Arabidopsis. thaliana* seeds were germinated on ½ MS plates supplemented with 1% sucrose. 10-d-old seedlings were transferred to soil and grown under short day conditions (8 hr light/16 hr dark).

## Cloning

REM1.2, CPK3, and CPK3[CA] sequences were previously published (*Perraki et al., 2018*). CPK3[K107M] was generated by site-directed mutagenesis using CPK3 as a template and primers specified in the *Supplementary file 1*.

All vectors built for this project, except for CRISPR and some protein production, were generated using multisite Gateway cloning strategies (http://www.lifetechnologies.com/) with pDONR P4-P1r, pDONR P2R-P3, pDONR221 as entry vectors. pLOK180_pR7m34g (*Noack et al., 2022*) was used as a destination vector for plant expression. pGEX-2T (GE Healthcare, N-terminal fusion with GST) was used for CPK protein production in bacteria for previously reported constructs (*Boudsocq et al., 2012*) (CPK2/3/5/11). CPK1, CPK3[K107M] and CPK6 were cloned into pGEX-3X-GW using the gateway cloning system and pDONR207 as entry vector. The N-terminal 118 amino acids of REM1.2 (REM1.2[1-118]) was synthetized with optimized codons for bacterial expression by GenScript (genscript.com) and cloned into pET24a (C-terminal fusion with a 6-histidine tag) between Nde1 and XhoI.

To generate CRISPR lines, sgRNAs targeting the N-terminus of *CPK3* gene were selected using CRISPR-P 2.0 website (*Boudsocq et al., 2012*; http://crispr.hzau.edu.cn/CRISPR2) and cloned into pHEE401 backbone (*Wang et al., 2015*) carrying the gene coding for the zCas9 enzyme under the control of Egg cell promoter using the Golden Gate cloning method.

All constructs were propagated using the NEB10 *E. coli* strain (New England Biolabs). Primers used for cloning are detailed in *Supplementary file 1*.

## Plant lines generation

T-DNA insertion mutants *rem1.2* (salk_117637.50.50 .x), *rem1.3* (salk_117448.53.95 .x) and *rem1.4* (SALK_073841.47.35) were provided by the ABRC. *rem1.2 rem1.3* double mutant was generated by crossing the respective T-DNA inserted parental plants, *rem2/rem1.3/rem1.4* was created by crossing *rem1.2 rem1.3* with *rem1.4*. The *cpk5 cpk6* (sail_657_C06, salk_025460) and *cpk5 cpk6 cpk11* (sail_657_C06, salk_025460, salk_054495) were described previously (*Boudsocq et al., 2010*). The quadruple mutant *cpk3 cpk5 cpk6 cpk11* was described previously (*Guzel Deger et al., 2015*). *cpk1 cpk2* (salk_096452, salk_059237) and *cpk1 cpk2 cpk5 cpk6* (salk_096452, salk_059237, sail_657_C06, salk_025460) were described previously (*Gao et al., 2013*). CPK3 T-DNA insertion lines *cpk3-1* (salk_107620) and *cpk3-2* (salk_022862) were obtained from Julian Schroeder (*Mori et al., 2006*) and Bernhard Wurzinger (*Mehlmer et al., 2010*), respectively. All mutants are in the same genetic ecotype Columbia Col-0. All plants were genotyped using primers indicated in the *Supplementary file 1*.

Pro35S:CPK3-HA #16.2, *cpk3-2*/Pro35S:CPK3-myc and *cpk3-2*/Pro35S:CPK3$^{K107M}$-myc used for viral propagation were previously published (*Lu et al., 2020*; *Concordet and Haeussler, 2018*). Pro35S:CPK3-HA #8.2 was generated at the same time as Pro35S:CPK3-HA #16.2, and protein expression was confirmed (*Figure 1—figure supplement 3*).

Col-0 plants were floral dipped with either ProUbi10:CPK3-mRFP1.2, ProUbi10:CPK3-mEOS3.2 or ProUbi10:mEOS3.2-REM1.2. *cpk3-2* plants were floral dipped with either ProUbi10:CPK3-mRFP1.2, ProUbi10:CPK3$^{G2A}$-mRFP1.2 or ProUbi10:mEOS3.2-REM1.2. *rem1.2 rem1.3 rem1.4* plants were floral dipped with ProUbi10:CPK3-mEOS3.2. Col-0/ProREM1.2:YFP-REM1.2 (*Rohr et al., 2024*) was transformed with ProUbi10:CPK3-mTagBFP2. Transformed seeds were selected based on the seedcoat RFP fluorescence.

For CRISPR-mediated site mutagenesis of CPK3, *rem1.2 rem1.3 rem1.4* was transformed by floral dip with the plasmid carrying z*Cas9* encoding gene and the sgRNA. Transformed candidates were selected on hygromycin and grown for seed collection. Harvested seeds were grown and a leaf sample was harvested for genomic DNA extraction. PCR were performed to amplify the targeted region, and CRISPR-induced mutations were screened using capillary electrophoresis. Mutated candidates were sent to sequencing to obtain homozygous lines for the mutation and then backcrossed with *rem1.2 rem1.3 rem1.4* to remove the Cas9. Primers used for CRISPR screening are listed in *Supplementary file 1*.

## Local viral propagation assay

Viral propagation assays were performed using PlAMV-GFP, an agroinfiltrable GFP-tagged infectious clone of PlAMV (*Minato et al., 2014*). *Agrobacterium tumefaciens* strain GV3101 carrying PlAMV-GFP was infiltrated on 3-wk-old *A. thaliana* plants at $OD_{600nm}$ = 0.2. Viral spreading was tracked using Axiozoom V16 macroscope system 5 d after infection. Infection foci were automatically analyzed using the Fiji software (http://www.fiji.sc/) via a homemade macro. The statistical significance was assessed using a two-way ANOVA, followed by a Tukey's multiple comparison test.

## Systemic viral propagation assay

At 3-wk-old, leaves were infiltrated with *Agrobacterium tumefaciens* GV3101 strain carrying PlAMV-GFP vector. Infection was followed every 3-4 d from the 10[th] day of infection to the 17[th] using a closed GFP-CAM FX 800-0/1010 GFP camera and the Fluorcam7 software (Photon System Instruments, Czech Republic; https://fluorcams.psi.cz/). Image analysis was performed using Fiji software (https://fiji.sc/). Integrated density of the fluorescence of systemic leaves was measured. Two independent experiments with at least 30 plants per genotype were performed. Statistical significance was determined with a Mann-Whitney test or a Kruskal-Wallis followed by a Dunn's multiple comparison test, depending on the number of conditions.

## Confocal microscopy

Live imaging was performed using a Zeiss LSM 880 confocal laser scanning microscopy system using either an oil-immersion PL-APO 40 x or 68 x (NA = 1.4) objective coupled with an AiryScan detector

for the latter. mRFP1.2 fluorescence was observed using an excitation wavelength of 561 nm and an emission wavelength of 579 nm. Acquisition parameters remained the same across experiments for SCI quantification. The SCI was calculated as previously described *Gronnier et al., 2017*. Briefly, 10 µm lines were plotted across the samples and the SCI was calculated by dividing the mean of the 5% highest values by the mean of 5% lowest values. Three lines were randomly plotted per cell. Three independent experiments were done, on at least 10 cells each time. Fenpropimorph (10 µg/mL) or DMSO was infiltrated 24 hr before observation.

## spt-PALM microscopy

*N. benthamiana* and *A. thaliana* epidermal cells were imaged at RT. Samples of leaves of 3-wk-old plants stably or transiently expressing mEOS3.2-tagged constructs were mounted between a glass slide and a cover slip in a drop of water to avoid dehydration. With the exception of ProUbi10:mEOS3.2-REM1.2 transiently expressed in *N. benthamiana*, image acquisitions were done on an inverted motorized microscope Nikon Ti Eclipse equipped with a 100 Å~oil-immersion PL-APO objective (NA = 1.49), a TIRF arm and a sCMOS Camera FUsion BT (Hamamatsu). Laser angle was adjusted to obtain the highest signal-to-noise ratio while laser power of a 405 nm and 561 nm laser, respectively to activate and image mEOS3.2, was adjusted to obtain a sufficient concentration of individual particles. Particle localization, tracks reconstructions, diffusion coefficient, and MSD parameters were obtained using PALMtracer, as previously described (*Gronnier et al., 2017*). The diffusion coefficient was calculated from the four first points of the MSD. Three independent experiments were conducted for each tested condition. Tessellation analysis was conducted using SR-Tesseler as previously described (*Levet et al., 2015*; *Gronnier et al., 2017*). ND were considered to be at least 32 nm², to contain at least five particles, and to have a particle density twice the average particle density within the sample.

For ProUbi10:mEOS3.2-REM1.2 transiently expressed in *N. benthamiana* in the presence or absence of ProUbi10:CPK3[CA]-mVenus or ProUbi10:CPK3[CAG2A]-mVenus, images were acquired on a custom-built platform, as described previously (*Rohr et al., 2024*). Particle localization, tracks reconstructions, diffusion coefficient and MSD parameters were obtained using OneFlowTraX (*Rohr et al., 2024*). Particle localization was done using the following parameters: photon/ADU conversion of 0.48, an offset of 100, a pixel size of 100 nm, a filter size of 1.2, a cut-off of 3, and a PSF of 7. Track reconstruction was done with a maximum linking distance of 200 nm, a maximum gap frame of 4. Tracks smaller than 8 localizations were filtered out and the diffusion coefficient was calculated from the second to fifth data points.

## RT-qPCR

3-wk-old *A. thaliana* leaves were infiltrated with PlAMV-GFP at final $OD_{600\,nm}$ = 0.2. Leaf samples were harvested 7 d after infiltration and immediately frozen. RNA extraction was done using Qiagen Plant Mini Kit and was followed by a DNase treatment. cDNA was produced from the extracted RNA using Superscript II enzyme from Invitrogen. RT-qPCR was performed on obtained cDNA using the iQ SYBR Green supermix (BioRad) on the iQ iCycler thermocycler (BioRad). The transcript abundance in samples was determined using a comparative threshold cycle method and was normalized to actin expression. Statistical differences were determined using a Mann-Whitney test. Primers used for RT-qPCR are listed in *Supplementary file 1*.

## Production of recombinant proteins and in vitro kinase assay

GST-CPK proteins were produced in BL21(DE3)pLys and purified as previously reported (*Concordet and Haeussler, 2018*). 6His-AtREM1.2[1-118] was produced like GST-CPK and purified on Protino Ni-TED column following themanufacturer's instructions (Macherey-Nagel). After elution with 40–250 mM imidazole, proteins were dialyzed overnight in 30 mM Tris HCl pH 7.5, 10% glycerol. Kinase assay was performed as previously described (*Perraki et al., 2018*) using 400 ng recombinant GST-CPK and 1–2 µg substrate (6His-AtREM1.2[1-118] or histone).

## Production of CPK3 antibodies

Polyclonal antibodies against *Arabidopsis* CPK3 were raised in rabbits by Covalab (France) using purified recombinant GST-CPK3. The antibodies were purified from rabbit serum by affinity chromatography on CH-Sepharose 4B (GE Healthcare) coupled to 6His-CPK3.

To produce the recombinant 6His-CPK3 and GST-CPK3 proteins, the *Arabidopsis* CPK3 cDNA was cloned into the expression vectors pDEST17 and pDEST15 (Invitrogen), respectively. Expression of 6His-CPK3 was induced in *Escherichia coli* strain BL21-AI (Invitrogen) with 0.2% (m/v) arabinose and the recombinant protein was affinity purified using Ni-NTA agarose (Qiagen). For GST-CPK3, protein expression in *E. coli* Rosetta cells (Novagen) was induced with 0.5 mM IPTG (isopropyl-β-D-thiogalactopyranoside) and recombinant GST-CPK3 was purified by Glutathione Sepharose 4 Fast Flow chromatography (GE Healthcare) as described by the manufacturer.

## Western blots

Protein samples were extracted from *A. thaliana* leaf tissue using 2 X Laemmli buffer or in a buffer containing 50 mM Tris HCl pH 7.5, 5 mM EDTA, 5 mM EGTA, 1 X anti-protease cocktail [Roche], 1% Triton X-100, 2 mM DTT. Proteins were transferred to PVDF and detected with polyclonal antibodies raised against CPK3 or REM1.2/REM1.3 (*Grison et al., 2019*), followed by incubation with secondary anti-rabbit HRP-conjugated antibodies (Sigma, #12–348).

## Material availability statement

All plasmids and *Arabidopsis* lines newly created in the frame of this work is available upon request to the corresponding author.

## Acknowledgements

We thank Thierry Mauduit and Christophe Higelin (HPE Greenhouse, INRAe) for plant culture. We thank the Bordeaux Imaging Center, part of the National Infrastructure France-BioImaging supported by the French National Research Agency (ANR-10-INBS-04). This work was supported by the French National Research Agency (grant no. ANR-19-CE13-0021 to SGR, SM, VG, MB) and the German Research Foundation (DFG) grant CRC1101-A09 to JG, the IPS2 benefits from the support of the LabEx Saclay Plant Sciences-SPS (ANR-10-LABX-0040-SPS). This study received financial support from the French government in the framework of the IdEX Bordeaux University 'Investments for the Future' program/GPR 'Bordeaux Plant Sciences.'

## Additional information

### Funding

| Funder | Grant reference number | Author |
| --- | --- | --- |
| Agence Nationale de la Recherche | ANR-10-INBS-04 | Sébastien Mongrand |
| Agence Nationale de la Recherche | ANR-19-CE13-0021 | Sébastien Mongrand |
| Deutsche Forschungsgemeinschaft | CRC1101-A09 | Julien Gronnier |
| Labex Saclay Plant Science | ANR-10-LABX-0040-SPS | Marie Boudsocq |
| Investments for the Future GPR | Bordeaux Plant Sciences | Véronique Germain |

The funders had no role in study design, data collection and interpretation, or the decision to submit the work for publication.

### Author contributions

Marie-Dominique Jolivet, Conceptualization, Resources, Data curation, Formal analysis, Validation, Investigation, Visualization, Methodology, Writing – original draft, Writing – review and editing; Anne Flore Deroubaix, Nikolaj B Abel, Marion Rocher, Dorian Lefebvre, Grégoire Saias, Jahed Ahmed, Data curation, Formal analysis; Marie Boudsocq, Conceptualization, Resources, Data curation, Formal analysis, Supervision, Validation, Visualization, Writing – original draft, Writing – review and editing; Terezinha Robbe, Data curation; Valérie Wattelet-Boyer, Conceptualization, Resources, Data

curation, Formal analysis; Jennifer Huard, Resources, Data curation; Yi-Ju Lu, Brad Day, Yasuyuki Yamaji, Resources; Valérie Cotelle, Resources, Data curation, Formal analysis; Nathalie Giovinazzo, Resources, Data curation, Formal analysis, Validation; Jean-Luc Gallois, Resources, Data curation, Formal analysis, Supervision; Sylvie German-Retana, Resources, Funding acquisition, Visualization, Writing – original draft, Project administration; Julien Gronnier, Resources, Data curation, Formal analysis, Writing – original draft; Thomas Ott, Resources, Data curation, Formal analysis, Funding acquisition, Writing – original draft; Sébastien Mongrand, Véronique Germain, Conceptualization, Resources, Data curation, Software, Formal analysis, Supervision, Funding acquisition, Validation, Investigation, Visualization, Methodology, Writing – original draft, Project administration, Writing – review and editing

### Author ORCIDs
Julien Gronnier ⓘD https://orcid.org/0000-0002-1429-0542
Thomas Ott ⓘD https://orcid.org/0000-0002-4494-9811
Sébastien Mongrand ⓘD https://orcid.org/0000-0002-9198-015X
Véronique Germain ⓘD https://orcid.org/0000-0001-6322-1204

Reviewer #1 (Public review): https://doi.org/10.7554/eLife.90309.3.sa1
Reviewer #3 (Public review): https://doi.org/10.7554/eLife.90309.3.sa2
Author response https://doi.org/10.7554/eLife.90309.3.sa3

## Additional files

### Supplementary files
Supplementary file 1. Primers used in this study.

MDAR checklist

### Data availability
Code availability statementAll R codes created in the frame of this work is available on Github (https://github.com/MD-Jolivet/eLife_Jolivet_2023; copy archived at *Jolivet, 2025*). Data availability statementAll raw images obtained in the frame of this work were deposited on BioImage Archive (https://www.ebi.ac.uk/biostudies/bioimages/studies/S-BIAD1733).

The following dataset was generated:

| Author(s) | Year | Dataset title | Dataset URL | Database and Identifier |
| --- | --- | --- | --- | --- |
| Jolivet MD | 2025 | Source data for "Interdependence of plasma membrane nanoscale dynamics of a kinase and its cognate substrate underlies Arabidopsis response to viral infection" | https://www.ebi.ac.uk/biostudies/bioimages/studies/S-BIAD173 | ArrayExpress, S-BIAD173 |

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
