## [Editor Report · eLife Assessment]

The study is considered **important** with **solid** evidence that demonstrates the impact of plasma membrane nano-domains and protein interactions in the plant defence response to viruses. It includes a molecular understanding of the role of a calcium dependent kinase (CPK3) and a remorin protein in the cell-to-cell spread of viruses and cytoskeletal dynamics demonstrating, conclusively, the role of CPK3 with multiple lines of evidence. The work opens avenues to investigate different viruses and other plasma membrane proteins to gain a fuller picture of the involvement of plasmodesmata and other nanodomains in virus spreading.

---

## [Referee Report · Reviewer #1 (Public review)]

Summary:

How plants perceive their environment and signal during growth and development is of fundamental importance for plant biology. Over the last few decades, nano domain organisation of proteins localised within the plasma-membrane has emerged as a way of organising proteins involved in signal pathways. Here, the authors addressed how a non-surface localised signal (viral infection) was resisted by PM localised signalling proteins and the effect of nano domain organisation during this process. This is valuable work as it describes how an intracellular process affects signalling at the PM where most previous work has focused on the other way round, PM signalling effecting downstream responses in the plant. They identify CPK3 as a specific calcium dependent protein kinase which is important for inhibiting viral spread. The authors then go on to show that CPK3 diffusion in the membrane is reduced after viral infection and study the interaction between CPK3 and the remorins, which are a group of scaffold proteins important in nano domain organisation. The authors conclude that there is an interdependence between CPK3 and remorins to control their dynamics during viral infection in plants.

Strengths:

The dissection of which CPK was involved in the viral propagation was masterful and very conclusive. Identifying CPK3 through knockout time course monitoring of viral movement was very convincing. The inclusion of overexpression, constitutively active and point mutation non-functioning lines further added to that.

Weaknesses:

I would like to thank the researchers for including some additional work suggested in the previous round of peer review. However, I still have concerns over this work which are two fold.

(1) Firstly, the imaging described and shown is not sufficient to support the claims made. The PM localisation and its non-PM localised form look similar and with no PM stain or marker construct used to support this. In addition, the quality of lots of the confocal based imaging (including new figure on colocalisation) is simply not sufficient. The images are too noisy and no clear conclusions can be made. The point made previously, the system this data was collected on has an Airyscan detector capable of 120nm resolution and as such NDs can be resolved. The sptPALM data conclusions are nice and fit the narrative. The inclusion of sptPALM movies is useful for the reader and tracks numbers is highly beneficial. But they do not show a high signal to noise ratio compared to other work in the field (see work from Alex Martineire) and the mEOS prticles are only just observable over the detector noise in some videos. As such, I worry about the data quality on which the analysis is based on. In addition, in some of the videos the conversion laser seems too high as it is difficult to separate some of the single particles as they emerge which would again, hinder the analysis.

(2) Secondly, remorins are involved in a lot of nano domain controlled processes at the PM. The authors have not conclusively demonstrated that during viral infection the remorin effects seen are solely due to its interaction with CPK3. The sptPALM imaging of REM1.2 in a cpk3 knockout line goes part way to solve this and the inclusion of CPK3-CA also strengthens the authors claims. But to propose a kiss and go model bearing in mind the differences in diffusion between CPK3 and REM3 and differential changes to diffusion between the two proteins after PIAMV infection without two colour imaging of both proteins at the same time, the claims are much stronger than the evidence. Negative control experiments are required here utilising other PM localised proteins which have no role during viral infection (such as Lti6B).

Overall, I think this work has the potential to be a very strong manuscript but additional evidence supporting interaction claims would significantly strengthen the work and make it exceptional.

---

## [Referee Report · Reviewer #3 (Public review)]

Summary:

This study examined the role that the activation and plasma membrane localisation of a calcium dependent protein kinase (CPK3) plays in plant defence against viruses.

The authors clearly demonstrate that the ability to hamper the cell-to-cell spread of the virus P1AMV is not common to other CPKs which have roles in defence against different types of pathogens, but appears to be specific to CPK3 in Arabidopsis. Further, they show that lateral diffusion of CPK3 in the plasma membrane is reduced upon P1AMV infection, with CPK3 likely present in nano-domains. This stabilisation however, depends on one of its phosphorylation substrates a Remorin scaffold protein REM1-2. However, when REM1-2 lateral diffusion was tracked, it showed an increase in movement in response to P1AMV infection. These contrary responses to P1AMV infection were further demonstrated to be interdependent, which led the authors to propose a model in which activated CPK3 is stabilised in nano-domains in part by its interaction with REM1.2, which it binds and phosphorylates, allowing REM1-2 to diffuse more dynamically within the membrane.

The likely impact of this work is that it will lead to closer examination of the formation of nano-domains in the plasma membrane and dissection of their role in immunity to viruses, as well as further investigation into the specific mechanisms by which CPK3 and REM1-2 inhibit the cell-to-cell spread of viruses, including examination of their roles in cytoskeletal dynamics.

Strengths:

The paper provided compelling evidence about the roles of CPK3 and REM1-2 through a combination of logical reverse genetics experiments and advanced microscopy techniques, particularly in single particle tracking.

Weaknesses:

There is limited discussion or exploration of the role that CPK3 has in cytoskeletal organisation and whether this may play a role in the plant's defence against viral propagation. Further. although the authors show that there is no accumulation of CPK3/Rem1.2 at plasmodesmata, it would be interesting to investigate whether the demonstrated reduction of viral propagation is due to changes in PD permeability.

---

## [Author Response]

The following is the authors’ response to the original reviews.

**Public Reviews:**

**Reviewer #1 (Public Review):**
Summary:How plants perceive their environment and signal during growth and development is of fundamental importance for plant biology. Over the last few decades, nano domain organisation of proteins localised within the plasma-membrane has emerged as a way of organising proteins involved in signal pathways. Here, the authors addressed how a non-surface localised signal (viral infection) was resisted by PM localised signalling proteins and the effect of nano domain organisation during this process. This is valuable work as it describes how an intracellular process affects signalling at the PM where most previous work has focused on the other way round, PM signalling effecting downstream responses in the plant. They identify CPK3 as a specific calcium dependent protein kinase which is important for inhibiting viral spread. The authors then go on to show that CPK3 diffusion in the membrane is reduced after viral infection and study the interaction between CPK3 and the remorins, which are a group of scaffold proteins important in nano domain organisation. The authors conclude that there is an interdependence between CPK3 and remorins to control their dynamics during viral infection in plants.Strengths:The dissection of which CPK was involved in the viral propagation was masterful and very conclusive. Identifying CPK3 through knockout time course monitoring of viral movement was very convincing. The inclusion of overexpression, constitutively active and point mutation non functioning lines further added to that.Weaknesses:My main concerns with the work are twofold.(1) Firstly, the imaging described and shown is not sufficient to support the claims made. The PM localisation and its non-PM localised form look similar and with no PM stain or marker construct used to support this. The sptPALM data conclusions are nice and fit the narrative. However, no raw data or movie is shown, only representative tracks. Therefore, the data quality cannot be verified and in addition, the reporting of number of single particle events visualised per experiment is absent, only number of cells imaged is reported. Therefore, it is impossible for the reader to appreciate the number of single molecule behaviours obtained and hence the quality of the data.(2) Secondly, remorins are involved in a lot of nanodomain controlled processes at the PM. The authors have not conclusively demonstrated that during viral infection the remorin effects seen are solely due to its interaction with CPK3. The sptPALM imaging of REM1.2 in a cpk3 knockout line goes part way to solve this but more evidence would strengthen it in my opinion. How do we not know that during viral infection the entire PM protein dynamics and organisation are altered? Or that CPK3 and REM are at very distant ends of a signalling cascade. Negative control experiments are required here utilising other PM localised proteins which have no role during viral infection. In addition, if the interaction is specific, the transiently expressed CPK3-CA construct (shown to from nano domains) should be expressed with REM1.2-mEOS to show the alterations in single particle behaviour occur due to specific activations of CPK3 and REM1.2 in the absence of PIAMV viral infection and it is not an artefact of whole PM changes in dynamics during viral infection.In addition, displaying more information throughout the manuscript (such as raw particle tracking movies and numbers of tracks measured) on the already generated data would strengthen the manuscript further.Overall, I think this work has the potential to be a very strong manuscript but additional reporting of methods and data are required and additional lines of evidence supporting interaction claims would significantly strengthen the work and make it exceptional.
**Reviewer #2 (Public Review):**
Summary:The paper provides evidence that CPK3 plays a role in plant virus infection, and reports that viral infection is accompanied by changes in the dynamics of CPK3 and REM1.2, the phosphorylation substrate of CPK3, in the plasma membrane. In addition, the dynamics of the two proteins in the PM are shown to be interdependent.Strengths:The paper contains novel, important information.Weaknesses:The interpretation of some experimental data is not justified, and the proposed model is not fully based on the available data.
**Reviewer #3 (Public Review):**
Summary:This study examined the role that the activation and plasma membrane localisation of a calcium dependent protein kinase (CPK3) plays in plant defence against viruses.The authors clearly demonstrate that the ability to hamper the cell-to-cell spread of the virus P1AMV is not common to other CPKs which have roles in defence against different types of pathogens, but appears to be specific to CPK3 in Arabidopsis. Further they show that lateral diffusion of CPK3 in the plasma membrane is reduced upon P1AMV infection, with CPK3 likely present in nano-domains. This stabilisation however, depends on one of its phosphorylation substrates a Remorin scaffold protein REM1-2. However, when REM1-2 lateral diffusion was tracked, it showed an increase in movement in response to P1AMV infection. These contrary responses to P1AMV infection were further demonstrated to be interdependent, which led the authors to propose a model in which activated CPK3 is stabilised in nano-domains in part by its interaction with REM1.2, which it binds and phosphorylates, allowing REM1-2 to diffuse more dynamically within the membrane.The likely impact of this work is that it will lead to closer examination of the formation of nano-domains in the plasma membrane and dissection of their role in immunity to viruses, as well as further investigation into the specific mechanisms by which CPK3 and REM1-2 inhibit the cell-to-cell spread of viruses.Strengths:The paper provided compelling evidence about the roles of CPK3 and REM1-2 through a combination of logical reverse genetics experiments and advanced microscopy techniques, particularly in single particle tracking.Weaknesses:There is a lack of evidence for the downstream pathways, specifically whether the role that CPK3 has in cytoskeletal organisation may play a role in the plant's defence against viral propagation. Also, there is limited discussion about the localisation of the nano-domains and whether there is any overlap with plasmodesmata, which as plant viruses utilise PD to move from cell to cell seems an obvious avenue to investigate.
**Recommendations for the authors:**

**Reviewer #1 (Recommendations For The Authors):**
Viral spread work in CPK mutants with time courses is beautiful!Regarding my public points on my issues with the imaging:- Figure 2A shows 'PM' localisation of CPK3 and 'non-PM' imaging of CPK3-G2A. The images are near identical both showing cell outlines and cytoplasmic strands. Here a PM marker (such as Lti6B) tagged with a different fluorophore or PM stain should be used in conjunction with surface views (such as in Figure 2C) to show it really is at the PM and the G2A line is not.

Impaired membrane localization of CPK3-G2A is documented in Mehlmer et al., 2010 using microsomal fractionation. Although Figure 2A main purpose is to show correct expression of the constructs in the lines used for PlAMV propagation (Figure 2B), we replaced the images with wider view pictures to be more representative of the subcellular localization of CPK3 and CPK3-G2A.

- Regarding Figure 2C, this is extremely noisy and PM heterogeneity is barely observable over the noise from the system (looking at the edges of surface imaged). You mention low resolution was an issue. I notice from the methods you have taken confocal images on an Zeiss 880 with Airyscan. These images must be confocal but If imaged with Airyscan the PM heterogeneity would be much clearer (see work from John Runions lab).

Indeed, these are tangential views images obtained by Zeiss 880 with Airyscan. Based on tessellation analysis (Figure 2H-J), CPK3 is rather homogeneously distributed and forms ND of around 70nm of diameter. Objects of such size cannot be resolved using pixel reassignment methods such as Airyscan. Note also that AtREM in our study are less heterogeneously distributed than what was described in the literature for StREM1.3.

- Regarding all sptPALM data. At least an example real data image and video is required otherwise the data can’t be assessed. The work of Alex Martiniere (sptPALM) or Alex Jonson (TIRF) all show raw data so the reader can appreciate the quality of the data. In addition, number of events (particles tracked) has to be shown in the figure legend, not just number of cells otherwise was one track performed per cell? Or 10,000? Obviously where the N sits in this range gives the reader more or less confidence of the data.

We agree and we added example videos of sptPALM experiments in the supplementary data, we also indicated the number of tracked particles in the figure legends.

- On a slight technical aside, how do you know the cells being imaged for sptPALM with PIAMV are actually infected with the virus? In Fig 2C you use a GFP tagged version but in sptPALM you use none tagged. I think a sentence in methods on this would help clarify.

PlAMV-GFP was used for spt-PALM experiment and cell infection was assessed during PALM experiment. This is now precised in the corresponding figures and methods.

- I also have a concern over some of the representative images showing the same things between different figures. Your clustering data in 3F looks very convincing. However, in Figure 2H the mock and PIAMV-GFP look very similar. How is Figure 3F so different for the same experiment? Especially considering the scale bars are the same for both figures. Same for CPK3-mRFP1.2 in Fig 2C and 3A, the same thing is being imaged, at the same scale (scale bars same size) but the images are extremely different.

Figure 2 data were generated using CPK3 stably expressed in *A. thaliana* while Figure 3 data were obtained upon transient over-expression of CPK3 in N. benthamiana. We do not have a clear explanation for such a difference in CPK3 PM behavior, it could lie on a different PM composition or actin organization between those two species, this point is now addressed in the discussion.

- Line 193&194 - you state that the CA CPK3 is reminiscent of the CPK3 upon PIAMV expression. I don't agree, while CPK3CA is less mobile (2D), the MSD shows it is in-between CPK3 and CPK3 + PIAMV. Therefore, can’t the opposite also be true? That overall the behaviour of CPK3-CA is reminiscent of WT CPK. I think this needs rewording.

We agree and we reworded that part

- Line 651 - what numerical aperture are you using for the lens during confocal microscopy. This is fundamentally important information directly related to the reproducibility of the work. You report it for the sptPALM.

The numerical aperture is now indicated in the methods.

Regarding my bigger point about specific interactions between CPK3 and remorin during viral infection to strengthen your claim the following need doing. I am not suggesting you do all of these but at least two would significantly enhance the paper.(1) Image a none related PM protein during viral infection using sptPALM and demonstrate that its behaviour is not altered (such as lti6b). This would show the affects on remorin behaviour are specific to CPK3 and not a whole scale PM alteration in dynamics due to viral infection.(2) Two colour SPT imaging of CPK3 and REM1.2. You show in absence of proteins (knockouts effect on each other) but your only interaction data is from a kinase assay (where CPK1 and 2 also interact, even though they are not localised at the same place) and colocalisation data (see below). A two colour SPT imaging experiment showing interaction and clustering of CPK3 and REM1.2 with each other and the change in their behaviours when viral infected and simultaneously imaged would address all of my concerns.- On another note, the co-localisation data (fig 5 sup 4) needs additional analysis. I would expect most PM proteins to show the results you show as the data is very noisy. In order to improve I would zoom in to fill the field of view and then determine correlation and also when one image is rotated 90 degrees (as described in Jarsch et al., plant cell) to enhance this work.(3) In the absence of viral infection, but presence of CPK3-CA, is sptPALM REM1.2 behaviour in the PM altered, if so then the interaction is specific and changes in remorin dynamics are not due to whole scale PM changes during viral infection and the manuscript substantially strengthened.(4) Building on from (3), if you have a CPK3 mutated with both CPK3-CA and G2A this would be constitutively active and non-PM localised and as such should not affect Remorin behaviour if your model is true, this would strengthen the case significantly but I appreciate is highly artificial and would need to be done transiently.

Regarding the first point, since the role of PM proteins involved in potexvirus infection is barely assessed, picking a non-related PM protein might be tricky. The data obtained with mEOS3.2-REM1.2 expressed in cpk3 null-mutant point towards a specific role of CPK3 in PlAMV-induced REM1.2 diffusion and not a general alteration of PM protein behavior.

Regarding the second point, we already reported the in vivo interaction between AtCPK3CA and AtREM1.2/AtREM1.3 by BiFC in N.benthamiana (Perraki et al 2018) and AtCPK3 was shown to co-IP with AtREM1.2 (Abel et al, 2021). While we agree on the relevance of doing dual color sptPALM with CPK3 and REM1.2, it is so far technically challenging and we would not be able to implement this in a timely manner. For the colocalization, although the whole cell is displayed in the figure, the analysis was performed on ROI to fill the field of analysis.

We agree with the relevance of adding the colocalization analysis of randomized images (mTagBFP2 channel rotated 90 degrees), this is now added to Figure 5 – supplement figure 5.

Finally, for the third and fourth points, spt-PALM analysis of REM1.2 in presence of CPK3-CA and CPK3-CA-G2A was performed (Figure 5 - figure supplement 4). The results suggest a specific role of CPK3-CA in REM1.2 diffusion.

Minor points:Line 59 - from, I think you mean from.Line 63 - Reference needed after latter.Line 68 - Reference required after viral infection.Line 85 - Propose not proposed.Line 156 - Allowed us to not allows to.Line 204 - add we previously 'demonstrated'Line 622 and 623 - You say lines obtained from Thomas Ott. This is very odd phrasing considering he is an author. I appreciate citing the work producing the lines but maybe reword this

These points were corrected, thank you.

**Reviewer #2 (Recommendations For The Authors):**
The paper provides evidence that CPK3 plays a role in plant virus infection, and reports that viral infection is accompanied by changes in the dynamics of CPK3 and REM1.2, the phosphorylation substrate of CPK3, in the plasma membrane. In addition, the dynamics of the two proteins in the PM are shown to be interdependent. The paper contains novel, important information that can undoubtedly be published in eLife. However, I have some concerns that should be addressed before it can be accepted for publication.Major concernsWhen the authors say that CPK3 plays a role in viral propagation, it should be clarified what is meant by 'propagation', - replication of the viral genome, its cell-to-cell transport, or long-distance transport via the phloem. By default the readers will tend to assume the former meaning. In my opinion, the term 'propagation' is misleading and should be avoided.

We purposely chose the term “propagation” because it sums replication and cell-to-cell movement. Nevertheless, we previously showed that group 1 StREM1.3 doesn’t alter PVX replication (Raffaele et al., 2009 The Plant Cell). In this paper, as we do not investigate the role of AtREM1.2 or AtCPK3 in the replication of the viral PlAMV genome, we cannot state that these proteins are strictly involved in cell-to-cell movement of the virus.

The authors show that viral infection is associated with decreased diffusion of CPK3 and increased diffusion of REM1.2 in the PM. However, it remains unclear whether these changes are related to partial resistance to viral infection involving CPK3 and REM1.2, or whether they are simply a consequence of viral infection that may lead to altered PM properties and altered dynamics of PM-associated proteins. Therefore, the model presented in Fig. 6 appears to be entirely speculative, as it postulates that changes in CPK3 and REM1.2 dynamics are the cause of suppressed virus 'propagation'. In addition, the model implies that a decrease in CPK3 mobility leads to activation of its kinase activity. This view is not supported by experimental data (see my next comment). The model should be deleted (both as the figure and its description in the Discussion) or substantially reworked so that it finally relies on existing data.

For the first point, the results obtained from the additional experiments proposed by reviewer #1 supports the hypothesis of a direct impact of CPK3 on REM1.2 diffusion (Figure 5 - figure supplement 4).

We agree with the second point and reworked the model to remove the link between CPK3 activation and its increased diffusion.

The statement that 'changes in CPK3 dynamics upon PlAMV infection are linked to its activation' (line 194) is based on a flawed logic, and the conclusion in this section of Results ('changes in CPK3 dynamics upon PlAMV infection are linked to its activation') is incorrect, as it is not supported by experimental data. In fact, the authors show that CPK3 dynamics and clustering upon viral infection is somewhat reminiscent of the behavior of a CPK3 deletion mutant, which is a constitutively active protein kinase. However, this partial similarity cannot be taken as evidence that CPK3 dynamics upon PlAMV infection are related to its activation. Furthermore, the authors emphasize the similarity of the mutant and CPK3 in infected cells without taking into account a drastic difference in their localization (Fig. 3A, middle and right panels) showing that the reduced dynamics or the compared proteins may have different causes. I suggest the removal of the section 'CPK3 activation leads to its confinement in PM ND' from the paper, as the results included in this section are not directly related to other data presented.

The PM lateral organization of PM-bound CPKs in their native or constitutively active form as well as the role of lipid in such phenomenon was never shown before. We believe that this section contains relevant information for the community. We kept the section but reworded it to tamper the correlation made between CPK3 PM organization upon viral infection and its activation.

Line 270 - 'group 1 REMs might play a role in CPK3 domain stabilization upon viral infection'. This is an overstatement. The size of the CPK3-containing NDs may have no correlation with their stability.

We reworded the sentence.

Minor pointsLine 204 - we previously that Line 234 and hereafter - "the D" sounds strange. Suggest using "the diffusion coefficient".

This was reworded.

**Reviewer #3 (Recommendations For The Authors):**
The authors have previously demonstrated that there was an increase in REM1.2 localisation to plasmodesmata under viral challenge. It would be useful to see if there was any co-localisation of REM1.2 and CPK3 with plasmodesmata in response to PlAMV and how this is affected in the mutants. This could be carried out relatively simply using aniline blue.

These experiments were added to the supplementary data of Figure 2 – figure supplement 2. and Figure 4 – figure supplement 4. , no enrichment of CPK3 or REM1.2 at plasmodesmata could be observed upon PlAMV infection.

Fig 3 supplementary figure 2 would be better incorporated into the main body of Figure 3 as this underpins discussion on the involvement of lipids such as sterols in the formation of nanodomains.

We moved Figure 3 – Supplementary figure 2 to the main body of Figure 3.

Minor corrections:Whilst the paper is generally well written there are a number of grammatical errors:Line 1 & 2: Title doesn't quite read correctly, suggest a rewording for clarity.L31: Insert "a"after onlyL33: Replace "are playing" with "play"L34: Begin sentence "Viruses are intracellular pathogens and as such the role..."59: replace "form" with "from"L63: Insert "was demonstrated" after REM1.2L85: Replace "proposed" with "propose"L86: replace "encouraging to explore" with "which will encourage further exploration of "L129: replace "we'll focus on" with "we concentrated on"L131: insert "an" before ATPL138: change "among" to "amongst"L156: change "allows to analyse" to "allows the analysis of"L204: Insert "showed" after previously.L232: "root seedlings" should this be the roots of seedlings?L235: insert "to" after "as"L280: insert "a" after "only"L281: change " to play" with "as playing": change CA to superscriptL307: Insert "was" after "transcription"L320: change "display" to "displaying"L321: change "form" to forms"L340: "hampering" should come before viralL365: insert"us' after "allow"

Thank you, these were corrected